# ControlTac: Force- and Position-Controlled Tactile Data Augmentation with a Single Reference Image

## Abstract

Vision-based tactile sensing is widely used in perception, reconstruction, and robotic manipulation, yet collecting large-scale tactile data remains costly due to diverse sensor-object interactions and inconsistencies across sensor instances. Existing approaches to scaling tactile data—simulation and free-form tactile generation—often yield unrealistically rendered signals with poor transfer to highly dynamic real-world tasks. We propose ControlTac, a two-stage controllable framework that generates realistic tactile images conditioned on a single reference tactile image, contact force, and contact position. By grounding generation in these important physical priors, ControlTac produces realistic samples that effectively capture task-relevant variations. Across three downstream tasks and three real-world experiments, the augmented datasets using our approach consistently improve performance and demonstrate practical utility in dynamic real-world settings. Project page: https://controltac.github.io/

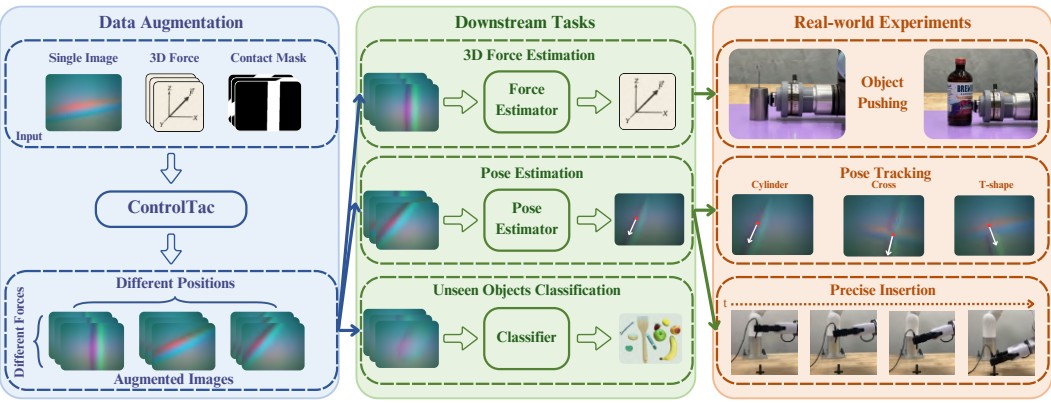

Figure 1: **Overview of ControlTac.** From a single reference image, ControlTac generates tens of thousands of realistic augmented tactile images conditioned on various contact forces and contact positions for a single contact surface on an object (left). The augmented tactile data improves multiple downstream tasks (middle) and transfers well to three dynamic real-world experiments (right).

## 1 Introduction

Vision-based tactile sensing has shown great promise for applications such as material classification (Li & Adelson, 2013; Gao et al., 2023b), 3D reconstruction (Huang et al., 2024b; Suresh et al., 2024; Swann et al., 2024), and robotic manipulation (Dong et al., 2021; Li et al., 2022; Yu et al., 2024). However, collecting large-scale tactile datasets remains costly because tactile sensing intrinsically requires physical contact across a wide range of positions and forces. Moreover, tactile images exhibit substantial variance arising from different gel properties and sensor lighting conditions, limiting cross-sample transfer and reuse. Together, these factors make tactile data scaling significantly more challenging than in vision and underscores the need for effective tactile data augmentation strategies.

| | Realism | Variation | Controllable |
|---|---|---|---|
| Text2Tac | Low | Low | ✗ |
| Vis2Tac | Low | Medium | ✗ |
| Simulation | Medium | Medium | ✓ |
| CONTROLTAC | High | High | ✓ |

Figure 2: **Comparison of tactile data augmentation approaches.** We evaluate whether each method produces visually realistic images, generates varied outputs from a single input (rather than collapsing to a mean image), and allows control via physical inputs. We compare CONTROLTAC with three other directions: Text2Tactile (Tu et al., 2025; Yang et al., 2024), Visual2Tactile (Li et al., 2019; Dou et al., 2024; Zhong et al., 2022), and Simulation (Wang et al., 2022; Si & Yuan, 2022; Si et al., 2024).

To scale up tactile datasets, two main approaches have been explored: *simulation-based methods* and *generative methods*. Simulation-based methods (Wang et al., 2022; Si & Yuan, 2022; Si et al., 2024) aim to render tactile images by modeling and simulating sensor-object interactions. However, they often suffer from inaccurate physics and rendering inconsistencies, resulting in unrealistic outputs that diverge from real tactile observations. Generative methods, on the other hand, use either text or visual input to generate the corresponding tactile image (Tu et al., 2025; Zhong et al., 2022; Dou et al., 2024; Yang et al., 2022; Gao et al., 2023a). While promising, these methods typically perform free-form generation without sufficient physical priors and can only generate one image for each contact, hindering the realism and usefulness of the resulting tactile data. As a result, both simulation and generative methods tend to produce data that poorly match real-world tactile signals, limiting their utility to pre-training (Zhao et al., 2024; Higuera et al., 2024; Gupta et al., 2025) or relative simple tasks such as contact localization (Gao et al., 2023b; Dou et al., 2024).

To harness generative methods while capturing wider real-world variability and cross-sensor differences, we argue that tactile data generation should be guided by both structured reference features and explicit physical constraints. Inspired by ControlNet (Zhang et al., 2023), which enhances visual generation by conditioning on edges, depth, and other cues, we propose to condition tactile generation on physical priors directly relevant to contact variation—contact force and contact position—and on a single real tactile image that supplies structural cues (e.g., contact geometry and sensor condition). This combination anchors the generator to the specific object-sensor instance and yields physically plausible tactile images that cover the variations required for downstream tasks.

To this end, we propose CONTROLTAC, a two-stage controllable tactile generation framework that generates realistic tactile images conditioned on a single reference tactile image, contact force, and contact position. In the first stage, the model takes as input a reference tactile image with an initial contact and a relative 3D force vector to generate a target image that reflects realistic deformation and texture under the specified force. In the second stage, a ControlNet-style architecture refines the force-controlled generation by incorporating a 2D contact mask, enabling precise control over contact position. With this design, CONTROLTAC can generate a large amount of physically plausible tactile images from *a single reference image*, extracting texture, geometry, and color cues while generalizing across objects and sensor instances by conditioning on varying contact forces and positions. As shown in Fig. 2, our framework outperforms alternative tactile data augmentation approaches by producing more realistic, diverse, and controllable tactile images.

We evaluate CONTROLTAC through extensive experiments and show that it can generate realistic, diverse tactile images from only a single reference image, significantly improving multiple downstream tasks—force estimation, contact pose estimation, and object classification—and proving effective in more challenging real-world dynamic manipulation tasks, such as object pushing and peg insertion. These gains arise because the generated data covers more variations in force, pose, and contact conditions that are difficult to capture exhaustively in fixed datasets. Moreover, CONTROLTAC generalizes well to previously unseen objects and shapes, as well as to different sensor instances.

Our contributions are threefold: (1) We propose a two-stage force and position controlled tactile generation framework for realistic tactile image generation and data augmentation; (2) We demonstrate that augmenting data with CONTROLTAC using only a single reference image significantly improves performance on three downstream tasks across unseen objects and sensor samples: force estimation,

pose estimation, and object classification; (3) We successfully deploy the models trained with our augmented data in real-world robot experiments, especially on the challenging object insertion task.

## 2 RELATED WORK

**Vision-based Tactile Sensing.** Various tactile sensors have been designed and used for different scenarios, including vision-based tactile sensors (Yuan et al., 2017a; Taylor et al., 2022; Lambeta et al., 2020; Lin et al., 2023), magnetic tactile sensors (Bhirangi et al., 2021; 2025), and piezo-resistive tactile sensors (Sundaram et al., 2019; Huang et al., 2024a). Among these, vision-based tactile sensors, which use a camera to capture RGB tactile images to record the deformations of gels when contact happens, have gained momentum due to their high spatial resolution and rich appearance cues. They have been used across a wide range of perception tasks, including liquid property classification (Huang et al., 2022), hardness classification (Yuan et al., 2017b), 3D reconstruction and generation (Huang et al., 2024b; Suresh et al., 2024; Swann et al., 2024; Gao et al., 2024; Shahidzadeh et al., 2024), slip detection (Li et al., 2018), and pose estimation (Huang et al., 2024b), as well as robotic manipulation tasks, including grasping (Calandra et al., 2018; 2017; Han et al., 2024), insertion (Yu et al., 2024; Li et al., 2022; Dong et al., 2021), pouring (Li et al., 2022), in-hand rotation (Qi et al., 2023), and dense packing (Yu et al., 2024; Dong & Rodriguez, 2019; Ai et al., 2024).

**Building Tactile Datasets.** To address the scarcity of tactile data, many prior works focus on collecting large-scale real-world datasets (Yang et al., 2022; Dou et al., 2024; Gao et al., 2021; Li et al., 2019). While these efforts help scale up tactile data, the quantity remains limited, and the resulting datasets are often difficult to reuse for downstream tasks—especially in robotics tasks—due to significant variability across sensors. Another approach is to use simulation (Wang et al., 2022; Si & Yuan, 2022; Si et al., 2024; Gao et al., 2022; Agarwal et al., 2021), which have been widely adopted for pre-training (Zhao et al., 2024; Higuera et al., 2024; Gupta et al., 2025; Feng et al., 2025) and Sim2Real transfer (Higuera et al., 2023; Gao et al., 2022; Si et al., 2024). However, bridging the Sim2Real gap remains a major challenge, as illustrated in Fig. 4. High-quality Sim2Real transfer typically still requires large real datasets for co-training (Wang et al., 2022) or the use of generative models for domain adaptation (Higuera et al., 2023). Differently, we introduce a controllable tactile generation framework to scale up tactile data under different physical conditions.

**Tactile Image Generation.** Text-to-tactile generation (Tu et al., 2025; Yang et al., 2024) and vision-to-tactile generation (Li et al., 2019; Dou et al., 2024; Yang et al., 2022; Zhong et al., 2022; Yang et al., 2023) have been widely used for representation learning (Feng et al., 2025; Zhao et al., 2024), contact localization (Dou et al., 2024; Gao et al., 2023b), classification (Dou et al., 2024; Zhong et al., 2022), and retrieval (Gao et al., 2023b; Yang et al., 2024). Cross-sensor generation (Rodriguez et al., 2024) has also been explored to utilize various properties of different tactile sensors. However, as shown in Fig. 2, the free-form generation from visual images often yields low-quality outputs. Moreover, only a single image can be generated from a contact location, which cannot cover the dynamic variations in real-world downstream tasks. To address this, we propose a conditional diffusion model that generates tactile images for data augmentation, guided by physical constraints and priors. We present both analysis and qualitative examples in Fig. 2 to highlight the limitations of existing approaches and the strengths of our method.

**Conditional Image Generation.** Conditional image generation has become a central topic in generative modeling, where the goal is to generate images guided by structured inputs such as class labels, text, or physical parameters. Early methods (Reed et al., 2016; Isola et al., 2017; Zhu et al., 2017; Bao et al., 2017; Tran et al., 2017; Yan et al., 2016) based on conditional GANs (Mirza & Osindero, 2014) and conditional VAEs (Sohn et al., 2015) demonstrate the feasibility of conditioning image generation on external inputs but often suffer from limitations in image quality and training stability (Karras et al., 2017; Fetaya et al., 2019; Jabbar et al., 2021; Oussidi & Elhassouny, 2018). More recently, diffusion models (Ho et al., 2020; Song et al., 2020a;b; Rombach et al., 2022; Chen et al., 2024; Xie et al., 2024) have emerged as state-of-the-art approaches due to their ability to generate high-fidelity and diverse images through a gradual denoising process. Meanwhile, ControlNet (Zhang et al., 2023; Gao et al., 2023a) enhances diffusion-based models by incorporating an auxiliary network that injects explicit structural conditions—such as edge maps, depth maps, or human poses—into the generation pipeline. This allows for fine-grained control over the output while maintaining the quality and diversity of diffusion models. Inspired by, but distinct from the prior work above, we tackle the new problem of controllable tactile image generation.

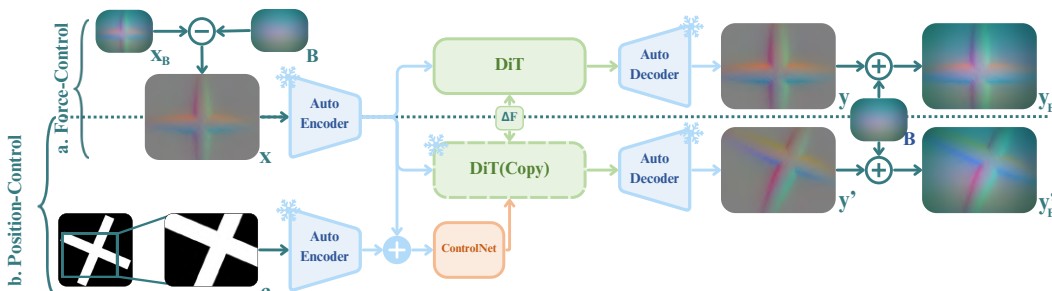

Figure 3: **Illustration of our controllable tactile generation framework. (a) Force-control stage:** given a background-subtracted tactile image $\mathbf{x} = \mathbf{x}_B - \mathbf{B}$ and a target relative 3D force $\Delta\mathbf{F}$, a conditional DiT generates a force-conditioned tactile image. **(b) Position-control stage:** we copy the force-control DiT and fine-tune it with a ControlNet conditioned on a contact mask $c$ to impose the desired 2D contact position (translation + rotation).

## 3 METHODOLOGY

We present CONTROLTAC, a controllable framework for generating realistic tactile images to scale up tactile data covering variations in downstream tasks, using only a single reference tactile image along with contact force and position. The key innovation lies in leveraging the reference tactile image to preserve contact geometry and color, while incorporating dynamic physical variations—force and contact pose—through a two-stage conditional tactile generation pipeline. This design ensures the generated tactile images are realistic and can cover real-world contact variations of downstream tasks.

In this section, we first introduce the architecture of our two-stage conditional tactile generation framework, which includes a force-control generator and a position-control generator (Sec. 3.2). Then, we demonstrate how to effectively leverage the generated data for downstream tasks such as force estimation, contact pose estimation, and object classification (Sec. 3.3). The overall architecture of our framework is illustrated in Fig. 3.

### 3.1 BACKGROUND OF VISUO-TACTILE SENSING

**Vision-based tactile sensors** (Yuan et al., 2017a; Lambeta et al., 2020; Taylor et al., 2022) are typically built from a soft elastomer gel, a textured background layer, an LED-based illumination module, and an onboard camera to observe the gel's deformation. The background layer is often embedded with markers to enhance contact force estimation (Taylor et al., 2022), while the illumination determines how shading and contact patterns are rendered. Although this design provides high-resolution tactile images, variations in gel stiffness, background fabrication, and lighting configurations can cause substantial appearance differences even between identical sensors (Zhao et al., 2024; Gupta et al., 2025). This problem limits cross-sensor data sharing and model transfer.

**Contact Force** from the tactile sensor is estimated and formulated via the deformation of the soft elastomer gel. In practice, recovering accurate contact force is highly non-trivial: the mapping from gel deformation to force depends on the gel's material properties (e.g., stiffness, thickness, damping), which often require finite-element modeling (FEM) for approximation (Ma et al., 2018). Moreover, a 3D contact force vector (with 1-D normal force and 2-D shear force) between the object and sensor does not correspond directly to the measured force distribution on sensor. Instead, it is entangled with local surface geometry, contact patch shape, and contact location on the object, which means that the same ground-truth force can result in different deformation patterns depending on how the object interacts. As a result, contact force estimation is inherently a challenging problem with vision-based tactile sensor (Shahidzadeh et al., 2025; Lin et al., 2023).

### 3.2 TWO-STAGE CONDITIONAL TACTILE GENERATION FRAMEWORK

We propose a two-stage conditional tactile image generation framework that incorporates force and contact position as controllable physical priors for covering the real-world dynamic variance in each contact. The model also leverages a reference tactile image to preserve color and texture cues. (1) In the first stage, the force-control generator takes a reference tactile image and the relative force as input to generate a target image that reflects the desired force. (2) In the second stage, we fine-tune the pretrained force-control generator with contact masks using ControlNet (Zhang et al., 2023) to control the contact position of generated tactile images.

**Force-Control Generation.** To enhance cross-sensor generation, we first preprocess the raw image $\mathbf{x_B}$ by subtracting the background $\mathbf{B}$, producing a background-removed reference image $\mathbf{x}$ that reduces sensor-specific variations such as lighting and emphasizes informative object–sensor contact features. This reference image is encoded via the frozen pretrained SANA (Xie et al., 2024) encoder $\mathcal{E}(\cdot)$ into a latent representation $\mathbf{z}^{(\mathbf{x})} = \mathcal{E}(\mathbf{x})$, preserving fine-grained texture and color information essential for characterizing object properties like hardness and surface roughness. The force-control generator $\mathcal{F}_f(\cdot)$ then takes $\mathbf{z}^{(\mathbf{x})}$ along with the relative force vector $\mathbf{\Delta F} = \mathbf{F}_t - \mathbf{F}_i$ as conditional inputs, performing conditional diffusion with the Diffusion Transformer (DiT (Peebles & Xie, 2023)) and DDIM (Song et al., 2020a) to generate the latent representation $\mathbf{z}^{(y)} = \mathcal{F}_f(\mathbf{z}^{(x)}, \mathbf{\Delta F})$ corresponding to the input representation $\mathbf{z}^{(\mathbf{x})}$ and related force $\Delta F$. Finally, the SANA decoder $\mathcal{D}(\cdot)$ reconstructs the target tactile image $\mathbf{y} \in \mathbb{R}^{W \times H \times 3}$, yielding improved cross-sensor generalization.

**Position-Control Mask.** We adopt a compact, object-specific binary template—the contact mask—as a global, object-level representation of the position-control signal. Each mask is computed once from a reference tactile image and remains fixed throughout the task, with changes in contact position modeled solely via rigid 2D transformations (translation + rotation). During training, masks are manually aligned with the ground truth with up to ±1 pixel/±1° precision, while at inference they can be automatically generated to any positions. This approach avoids common pitfalls in defining contact position: (1) it avoids the ambiguity of representing contact using a center point (x, y) and rotation angle, which becomes unreliable when the object is larger than the sensor's size; (2) it avoids the inconsistency of edge detection methods such as Canny (Canny, 1986), which can yield different contact boundaries depending on the contact force. In summary, the contact mask is a fixed, object-level template for position control, decoupled from force-induced local variations.

**Position-Control Generation.** We build on the pretrained force-control generator $\mathcal{F}_f(\cdot)$ and introduce a ControlNet (Zhang et al., 2023) structure to inject the contact mask as a control signal in covering more contact positions. Following PixArt-$\delta$ (Chen et al., 2024), ControlNet is applied to the first half of the DiT (Peebles & Xie, 2023) blocks. Each ControlNet block's output is added to the corresponding frozen block before passing to the next, enforcing the positional constraint. The generated tactile image $\mathbf{y}' = \mathcal{D}(\mathcal{F}_c(\mathbf{z}^{(\mathbf{x})}, \mathbf{z}^{(\mathbf{c})}, \mathbf{\Delta F}))$ satisfies both target force and contact position. Here, $\mathcal{F}_c(\cdot)$ is the force-control generator with ControlNet, and $\mathbf{z}^{(\mathbf{c})} = \mathcal{E}(\mathbf{c})$ is the latent contact mask $\mathbf{c} \in \mathbb{R}^{W \times H \times 1}$ encoded via SANA (Xie et al., 2024). This design enables fine-grained position control while preserving high-precision force control.

### 3.3 TACTILE DATA AUGMENTATION FOR DOWNSTREAM TASKS

To evaluate the effectiveness of data augmentation using CONTROLTAC, we evaluate three scenarios: tasks with force labels, tasks with pose labels, and tasks where labels remain unchanged after augmentation. Specifically, we select three tasks that require both realism and large-scale data to achieve effective augmentation: force estimation, contact pose estimation, and object classification.

**Force Estimation.** In this task, we adopt the force estimation framework proposed in FeelAny-Force (Shahidzadeh et al., 2025), which is based on a ViT (Dosovitskiy et al., 2020) encoder pretrained on DINOv2 (Oquab et al., 2023). The framework takes tactile images as input, consisting of a regressor that predicts the 3D force vector, which includes the normal force and the 2-directional shear force on the gel, and a decoder that reconstructs the depth image during training to enhance the training efficiency. It's notable to show that this task is highly sensitive to force, so each force label must correspond to an realistic tactile image, which is good for evaluating the realism of our generated images. More examples are shown in Fig. 13 of App. G.1. To our best knowledge, no prior work has attempted to use generated or simulated tactile images to tackle this highly challenging task.

**Contact Pose Estimation.** In this task, we also utilize the ViT (Dosovitskiy et al., 2020) encoder pretrained on DINOv2 (Oquab et al., 2023) from the force estimator in FeelAnyForce (Shahidzadeh et al., 2025). The regressor output is changed from the 3D force vector into the x and y coordinates of the center in the full contact mask, as well as the angle of the contact object relative to the tactile sensor. Additionally, the decoder's resconstrunciton supervision is shifted from depth image into the contact mask. Through these modifications, we obtain a pose estimator. More details of this task are shown in Sec. 4.3.

**Object Classification.** To ensure fair comparison, we use three common classifiers: a plain CNN, a ViT (Dosovitskiy et al., 2020) without pretraining, and a ViT pretrained on ImageNet (Deng et al.,

2009). The classification task involves six objects: five from the FeelAnyForce dataset (Shahidzadeh et al., 2025)—banana, marker, nectarine, ring, and thick cylinder—and one additional object collected by another sensor sample, the T-shape. Fig. 22 shows the objects and tactile images, and App. F provides more details on the classifiers.

## 4 EXPERIMENTS

In this section, we evaluate our proposed framework through extensive experiments in tactile image generation, data augmentation across three downstream tasks, and three real-world experiments. First, we evaluate the generation quality of our two-stage conditional tactile generation framework by comparing it with two baseline methods and a tactile simulation method in Sec. 4.1. Next, we evaluate the data augmentation capabilities of our framework through force estimation (Sec. 4.2), position estimation (Sec. 4.3), and object classification (Sec. 4.4). Finally, we deploy the trained force and pose estimators in three real-world experiments, as detailed in Sec. 4.5.

We train the force-control generator component using 20,000 tactile images with corresponding 3D force vectors from FeelAnyForce (Shahidzadeh et al., 2025). The ControlNet for position-control generation is trained using 7,000 tactile images, where each object contributes 300 unique contact positions, also paired with 3D force vectors from FeelAnyForce (Shahidzadeh et al., 2025). Further information regarding training and inference is available in Appendix A.1 and A.2, respectively.

### 4.1 GENERATION QUALITY EVALUATION

We evaluate CONTROLTAC's generation quality against three baselines: TAXIM[1] (Si & Yuan, 2022), a sample-based tactile simulator that uses Real2Sim sensor calibration to bridge the Sim2Real gap and represents the state of the art in vision-based tactile simulation (Nguyen et al., 2024; Schneider et al., 2025); and two ablations of our two-stage framework: (1) a HYBRID model that conditions diffusion jointly on force and position, and (2) a SEPARATE-Control pipeline, where the position-control generator is trained on images produced by a trained force-control model. Other than those baselines, visual-to-tactile generative frameworks (Li et al., 2019; Zhong et al., 2022; Dou et al., 2024; Gao et al., 2023a) produce a single image per contact, unlike CONTROLTAC, do not support millimeter-level position control or fine-grained force control. We therefore exclude them from our comparisons here; illustrative anaylsis and examples are provided in App. D.1.

For training, the HYBRID model uses 7,000 samples; the SEPARATE-Control pipeline uses 20,000 samples for force control and 7,000 for position control. We evaluate using the SSIM and pixel-wise MSE metrics on test data from FeelAnyForce (Shahidzadeh et al., 2025), where six of them are the same as the training set but with different contact positions and forces, and two of them are unseen objects: nectarine and banana. Further details are in App. B.

Table 1 and Fig. 4 present the quantitative and qualitative comparisons, respectively. CONTROLTAC significantly outperforms TAXIM on both MSE and SSIM, indicating more realistic tactile image synthesis under comparable physical conditions. CONTROLTAC also surpasses our ablation baselines, validating the benefit of our two-stage design: in the HYBRID variant, the performance is constrained by the total data scarcity for force control, while in the SEPARATE-Control pipeline, errors from these two stages compound,

Table 1: Comparison in MSE↓ (lower is better) and SSIM↑ (higher is better). HYBRID represents Hybrid Force-Position Conditional Diffusion Model, SEPARATE represents Separate-Control Pipeline, and TAXIM represents the simulator.

| Method | MSE ↓ | SSIM ↑ |
|---|---|---|
| TAXIM (Si & Yuan, 2022) | 1054 | 0.68 |
| HYBRID | 31 | 0.81 |
| SEPARATE | 157 | 0.79 |
| **Ours** | 23 | 0.83 |

resulting in significantly worse overall performance. Moreover, the qualitative result in Fig. 4 and quantitative result in Table 7 demonstrates that the model can still generate high-quality tactile images for unseen objects, where the performance is similar to the seen objects.

To verify its robustenss across different reference images, we conduct experiments using ten reference images per object under identical force and position control show minimal variation. The quantitative

---

[1]Note that Taxim cannot directly simulate with force and position as inputs. To make a fair comparison, we manually add the position control and treat displacement in the z-direction as the normal force.

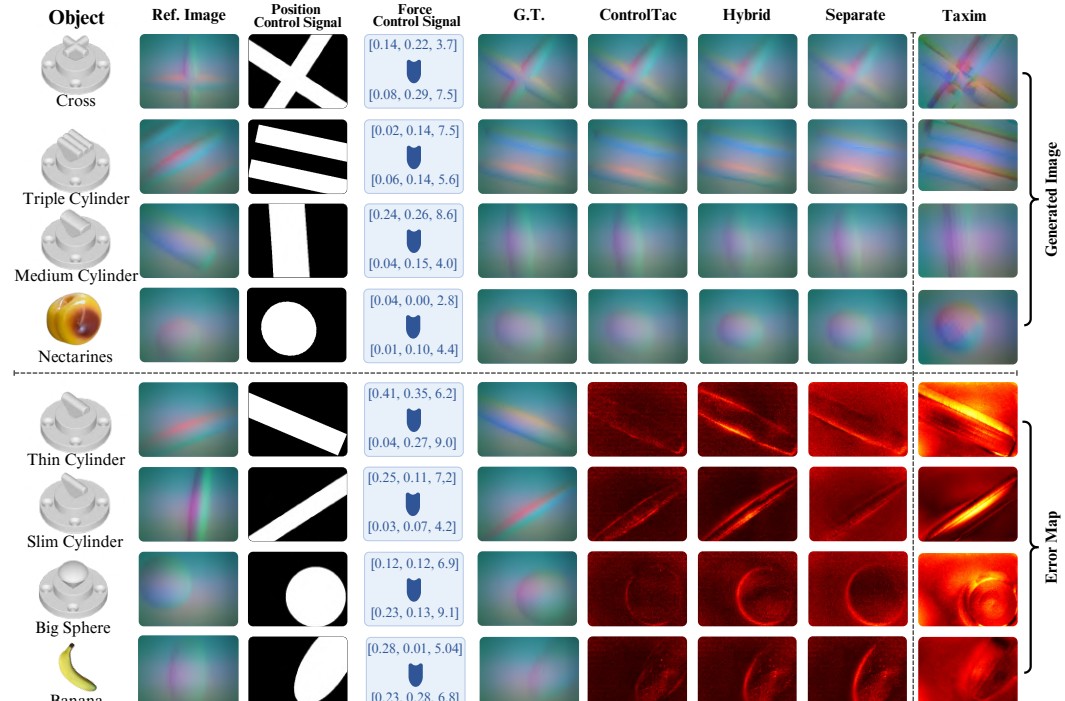

Figure 4: **Qualitative Generation Results.** The first column shows 3D previews of eight objects, followed by the input tactile image (Ref. Image) in the second column and the Contact Mask in the third column. The fourth column displays the initial force (top) and target force (bottom). Subsequent columns present the Ground Truth (G.T.) and results from CONTROLTAC, the hybrid force-position conditional diffusion model (Hybrid), the separate-control pipeline (Separate), and simulation results from Taxim (Si & Yuan, 2022). In the upper part, we visualize the generated images for comparison; in part lower part, we show the error maps highlighting differences from the ground-truth tactile image. Complete results are provided in Fig. 16 in appendix.

results has 95% confidence intervals of MSE (22.66-23.34) and SSIM (0.8249-0.8331), which suggest that the choice of reference images has negligible impact.

Additionally, since the RGB tactile images cannot fully capture the deformation of the gel and the corresponding force distribution, we also visualize the depth error distribution between the ground truth and the generated images. The depth visualizations in Fig. 17 and Fig. 18, along with error maps, further confirm that CONTROLTAC captures surface geometry rather than merely scaling brightness.

Additionally, to further validate the robustness of using contact masks, in App. C, we perturb the contact masks with controlled noise. Even with rotation error up to $6°$, translation error of 6 pixels, or scale changes of $0.8 \times /1.2\times$, the impact on the generated results is negligible. This indicates that errors in the contact mask caused by manual alignment are remaining independent of the contact area size, which reflects the applied force. Moreover, A detailed failure analysis is provided in App. E.

## 4.2 DOWNSTREAM TASK: FORCE ESTIMATION

Figure 5 demonstrates that CONTROLTAC can cover the variation of positions and force, and remarkably improves MAE even with small real subsets. With only a third of the real data, the performance can reach a comparable performance to the full dataset, where the performance with only a third real data is much worse since it cannot cover all the variations of forces and positions. It is worthy to note that combining all real + generated data performs slightly worse than using only real data, and this is because FeelAnyForce already achieves near-oracle performance with full forces and positions coverage, although it's challenging to collect them in the real world. App. D.4 analyzes the scaling capacity of the estimator with both real and generated data to address concerns about generation fidelity.

Given GelSight's high sensitivity to surface contact, estimating contact force has long been a promising research direction (Yuan et al., 2017a; Shahidzadeh et al., 2025). Even tiny force changes cause large image variations, making this task ideal for evaluating our method's generation quality. To validate that CONTROLTAC can generate realistic tactile images that cover more forces and positions, we use CONTROLTAC to generate 15,000 or 30,000 tactile images from a single reference image and co-train the model with (0-15k) real images to evaluate the performance. Specifically, the force range spans from 1 N to 10 N. For each contact position, we generate five randomly sampled force values with a precision of 0.1 N.

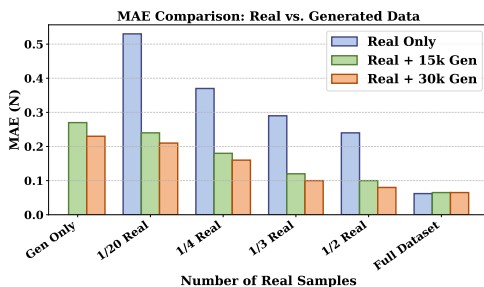

Figure 5: Force estimation performance (MAE) for different training sample sizes: using only generated data, 1/20 of the real data, 1/4 of the real data, 1/3 of the real data, 1/2 of the real data, and the full real dataset (15k samples).

We also perform ablation studies to highlight the importance in covering both forces and positions. In App. D.3, after covering all the positions, augmenting forces via the first stage of CONTROLTAC can significantly improve the performance. In App. G.3, the color distributions have significant differences with varied contact positions. Additionally, we show an example in App. G.2 to illustrate that CONTROLTAC can cover some contact positions where the real sensor cannot cover. Those results demonstrate the importance of both force and position augmentations.

### 4.3 DOWNSTREAM TASK: POSE ESTIMATION

We aim to demonstrate that the generated tactile images can cover more variations of the task and enhance its performance. We train four estimators: one for a cross, one for three cylinders (varying curvature/width), and two for unseen shapes (a T-shape and a Type-C connector, in Fig. 23). To test robustness across sensor instances, the T-shape and Type-C images come from a new instance (shown in Fig. 20). For all objects (cross, three-cylinder, T-shape, and Type-C connector), we used CONTROLTAC to generate training tactile data at various random contact positions and force, covering regions not covered in real data. For the seen cylinders and cross, we label some real data to compare with generated data for showing its effectiveness. The test sets consist of 30 annotated contact positions per object with varying force levels.

As shown in Table 2, pose estimators trained solely on tactile images generated by CONTROLTAC achieve strong

Table 2: Pose estimation errors (in pixels↓ and degrees↓ (lower is better)).

| Training Set | X ↓ | Y ↓ | Angle ↓ |
|---|---|---|---|
| *Cylinder (3 Types)* | | | |
| PCA (She et al., 2021) | 15 | 13 | 22 |
| Real | 8 | 8 | 4 |
| Sim (Si & Yuan, 2022) | 17 | 15 | 6 |
| Sim + Real | 12 | 13 | 6 |
| Gen (fixed) | 9 | 8 | 5 |
| **Gen (unfixed)** | **4** | **5** | **3** |
| **Gen (unfixed) + Real** | **3** | **4** | **3** |
| *Cross* | | | |
| PCA (She et al., 2021) | 56 | 19 | 18 |
| Real | 6 | 6 | 2 |
| Sim (Si & Yuan, 2022) | 19 | 18 | 5 |
| Sim + Real | 17 | 16 | 4 |
| Gen (fixed) | 7 | 9 | 4 |
| **Gen (unfixed)** | **3** | **4** | **1** |
| **Gen (unfixed) + Real** | **2** | **4** | **1** |
| *T-shape (Unseen)* | | | |
| Gen (unfixed) | 4 | 5 | 2 |
| *USB (Unseen)* | | | |
| Gen (unfixed) | 5 | 4 | 3 |

performance across all objects, including the unseen T Shape and USB with the new sensor sample. Remarkably, using the same amount of generated data outperforms training on real data alone, even when the real dataset is relatively large, as capturing tactile data that fully covers all contact variations in the dynamic real world is extremely challenging. In such case, generated data proves particularly valuable since all the covered positions can be generated.

Furthermore, CONTROLTAC not only outperforms simulation-based data from Taxim (Si & Yuan, 2022), where simulated images are not realistic, but also surpasses traditional PCA-based pose estimation methods. We also evaluate the pose estimator under varying versus fixed forces (denoted

as "fixed" in Table 2 set to the median value of 6.5 N). Results show that unfixed force improves performance since it covers the force variations in the real-world scenarios. Full comparisons are in App. D.5, with Taxim (Si & Yuan, 2022) visualized results in Fig. 21.

2D pose estimation has been used for cable insertion (She et al., 2021) and active perception (Ota et al., 2023). Unlike 3D pose estimators (Bauza et al., 2023), it does not require known object geometry, making it attractive when objects are unknown or vary across instances. Classical PCA-based 2D pose estimation has been used for cable manipulation (She et al., 2021), but it operates only on local contact area and often fails on asymmetric or large object. Learning-based 2D pose estimation (Wang et al., 2022) is more flexible but typically requires large, carefully labeled datasets, incurring substantial human effort. We also adopt the learning-based paradigm, but the training data can be automatically generated using our CONTROLTAC framework. Based on the full 2D contact surface masks for image generation, we can directly extract the labels from their centroids. Below, we compare with the PCA-based approach (She et al., 2021) and learning-based 2D pose estimators using/co-training with simulation-based data from Taxim (Si & Yuan, 2022).

### 4.4 DOWNSTREAM TASK: OBJECT CLASSIFICATION

To evaluate CONTROLTAC's generalizability against other augmentations, we perform an unseen objects classification task. Objects and tactile images are shown in Fig. 22.

In this experiment, we select one reference tactile image per object (six objects with two sensor samples) and use CONTROLTAC to generate images under varying force and contact positions. For comparison, a traditional augmentation pipeline (Maus et al., 2022; Yan et al., 2023) applies geometric transformations—rotations (eight 45° intervals), flipping (vertical, horizontal, both), scaling (0.8, 1.0, 1.2), and translations ([-20, 0, 20] along two axes), and also utilizes color transformations (hue shifts) to generate 6 variants, totaling 5,184 augmented images.

Table 3: Accuracy comparisons across models and augmentation methods. Geo: geometric data augmentation; Col: color augmentation; Gen: our CONTROLTAC-based augmentation method.

|                 | Geo  | Geo + Col | Gen  |
|-----------------|------|-----------|------|
| CNN             | 0.68 | 0.69      | **0.87** |
| ViT (Scratch)   | 0.60 | 0.65      | **0.95** |
| ViT (ImageNet)  | 0.76 | 0.79      | **0.99** |

We evaluate classification performance using three different models: a CNN, a ViT from scratch, and a ViT with pre-training. Detailed information about the classifiers can be found in App. F. We train the models with data samples of 4,800 using three augmentation methods. The results are summarized in Table 3. Across all models, CONTROLTAC consistently outperforms traditional augmentation methods, with especially notable improvements in the ViT-based models. This demonstrates the superior utility of CONTROLTAC in enhancing downstream classification performance.

### 4.5 CASE STUDY: REAL-WORLD EXPERIMENTS

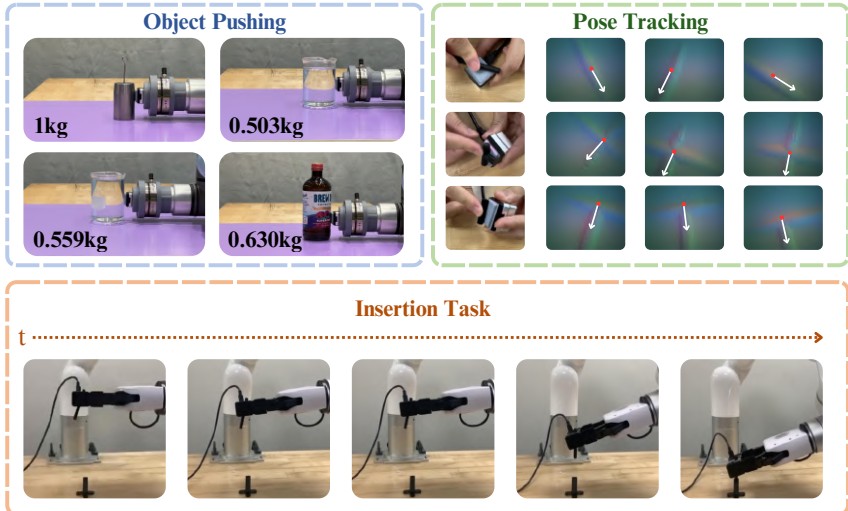

Figure 6: Qualitative examples of real-world experiments where we utilize the force and pose estimators trained with the augmented dataset using our method. See Supp. video for demos.

In this section, we utilize the force estimator and pose estimator by training with augmented dataset in three real-world experiments: Object Pushing, Real-time Pose Tracking, and Precise Insertion.

**Object Pushing.** In this experiment, we estimate the pushing force between the robot and four objects: a 1 kg calibration weight made of metal, a caliber cylinder full of water (0.559 kg) or almost full (0.503 kg), and a glass bottle of 0.63 kg. We utilize a UR5 robot with a ATI Axia80 force sensor to collect the ground truth forces for evaluations. Five pushes of approximately 15s each object are conducted. The details of experiment setting is shown in Fig. 6.

For evaluation, we compare our model trained on the dataset augmented by the force-control generator with the model trained on real data. As shown in Table 9, the key finding is that the force estimator trained on generated images achieves similar measurement error, within $0.1$ N, to the one trained on the real dataset, highlighting that it generalizes well to complex real-world scenarios and new objects with diverse textures, materials, and weights.

**Real-time Pose Tracking.** To evaluate the performance of our pose estimator, we conduct a real-time pose tracking experiment. Specifically, we press the printed cylinder, cross, and T-shape object into the sensor and change the object pose by rotating and translating. In this setting, our model can track the pose in real time with 10 Hz, which highlights the practicality of the model trained with our augmented data in this dynamic real-world scenario. A visualization of the task is shown in Fig. 6.

**Insertion Task.** For the insertion task, we 3D print three different objects (a cylinder, a cross-shaped object, a T-shape object) and a hole. Also, we set up a Type-C connector insertion task for inserting the USB into the charger. We utilize the XArm7 robot arm with two GelSight Mini tactile sensors to accomplish the task using our trained pose estimator. Notably, the tolerance of this insertion task is only 3 mm. See App. D.7 for more details about the task settings as well as a discussion of the ethics and safety concerns.

To evaluate the performance of our model, we conduct 20 insertion trials for each object. Our insertion strategy achieves a success rate of $90\%$ for the cylinder and $85\%$ for both the cross and T-shape objects. These results highlight the practicality of our augmented data, demonstrating that a model trained with data augmented from a single reference image can achieve strong performance on a challenging dynamic real-world insertion task with *only 3 mm of tolerance*. We further evaluated Type-C connector insertion by training a pose estimator with generated tactile images, achieving an impressive $75\%$ success rate. This demonstrates the effectiveness of our approach in daily object. The real object and corresponding tactile images of the Type-C connector are provided in Fig. 23.

## 5 CONCLUSION

We presented CONTROLTAC, a two-stage conditional tactile generation framework capable of generating realistic and diverse tactile images conditioned on varying forces and contact positions, all from a single reference tactile image. Through experiments on three downstream tasks and three real-world experiments, we demonstrate that CONTROLTAC enables effective data augmentation and performs well in practical and challenging real-world applications by capturing a wider range of dynamic, real-world variations. While our work marks the first attempt to enable controllable tactile image generation for data augmentation in dynamic downstream tasks, it currently cannot be paired with robot action data and visual observations. As future work, we plan to incorporate this framework with robot data to tackle more challenging contact-rich robot manipulations.

## 6 ETHICS STATEMENT

We have carefully reviewed the ICLR Code of Ethics. Our paper does not involve human subjects, sensitive personal data, or any experiments that may raise ethical concerns. It does not pose risks of harm, discrimination, privacy violations, or security issues. The datasets and methods used are publicly available and widely adopted in the research community. We are not aware of any conflicts of interest, legal compliance issues, or other ethical concerns related to this work.

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

APPENDIX OVERVIEW

This appendix provides comprehensive supplementary materials to support the main paper. We first present detailed implementation specifics, including model architectures, training configurations, loss functions, and evaluation metrics (Sec. A). Next, we describe the baseline methods employed for comparison, encompassing both hybrid and separate force-position conditional diffusion models (Sec. B).

We then report additional experimental results, including robustness analyses of contact mask alignment (Sec. C), an additional evaluation of generation quality (Sec. D.2), the influence of data composition on model performance (Sec. D.4), a complete comparison of pose estimation results across diverse objects (Sec. D.5), the object-pushing experimental results (Sec. D.6), as well as the experimental setup for precise insertion tasks in real-world robotic scenarios (Sec. D.7). Failure cases and limitations arising from dataset biases are discussed in Sec. E.

Furthermore, we provide detailed descriptions of the classifier architectures used in our study, including both CNN and Vision Transformer models (Sec. F) Finally, we include extensive visualizations to illustrate object orientations, error maps, depth maps, generated tactile images, simulated tactile data, classification examples, and the Type-C connector insertion task (Sec. G).

Collectively, this appendix offers rigorous methodological details, quantitative analyses, and qualitative illustrations to substantiate and extend the findings presented in the main text.

## A IMPLEMENTATION DETAILS

### A.1 TRAINING CONFIGURATION

Both force-control generation component and position-control generation component are trained using the AdamW (Loshchilov & Hutter, 2017) optimizer and a cosine annealing learning rate scheduler. For the force-control generator, the learning rate is annealed from an initial value of $1 \times 10^{-4}$ to a final value of $1 \times 10^{-5}$. For the position-control generator, the learning rate decays from $1 \times 10^{-5}$ to $1 \times 10^{-6}$. Each model is trained for 75,000 steps on a single NVIDIA RTX A5000 GPU with a batch size of 4. The loss function used for training is a weighted combination of L1 loss and mean squared error (MSE): $0.5 \times \mathcal{L}_{L1} + 0.5 \times \mathcal{L}_{MSE}$.

### A.2 INFERENCE CONFIGURATION

We perform inference on a single NVIDIA RTX A6000 GPU with a batch size of 128. ControlTac achieves a throughput of 6.5 tactile images per second, while the Hybrid and Separate baseline methods reach 7 and 3.7 tactile images per second, respectively. Since the VAE decoding process is the most memory-intensive step, all three methods exhibit a peak GPU memory usage of 29.97 GB.

### A.3 METRICS

We evaluate our models using several commonly used metrics, including mean squared error (MSE), L1 loss, mean absolute error (MAE), and structural similarity index measure (SSIM). Specifically, the following metrics are reported:

- **Mean Squared Error (MSE):** $\text{MSE} = \frac{1}{n} \sum_{i=1}^{n} (y_i - \hat{y}_i)^2$

- **L1 Loss:** $\text{L1} = \frac{1}{n} \sum_{i=1}^{n} |y_i - \hat{y}_i|$

- **Mean Absolute Error (MAE):** $\text{MAE} = \frac{1}{n} \sum_{i=1}^{n} |y_i - \hat{y}_i|$

- **Structural Similarity Index Measure (SSIM):** $\text{SSIM}(x, y) = \frac{(2\mu_x\mu_y + C_1)(2\sigma_{xy} + C_2)}{(\mu_x^2 + \mu_y^2 + C_1)(\sigma_x^2 + \sigma_y^2 + C_2)}$

# B    DETAILS OF BASELINES

## B.1    HYBRID FORCE-POSITION CONDITIONAL DIFFUSION MODEL:

In this approach, we train a diffusion model $\mathbf{y} = \mathcal{D}(\mathcal{F}_h(\mathbf{z}^{(\mathbf{x})}, \mathbf{z}^{(\mathbf{c})}, \mathbf{\Delta f}))$. Here, the latent representation of initial tactile image $\mathbf{z}^{(\mathbf{x})}$, contact mask $\mathbf{z}^{(\mathbf{c})}$, and target force change $\mathbf{\Delta f}$ are simultaneously input into the diffusion model $\mathcal{F}_h(\cdot)$, which is then passed into the autodecoder $\mathcal{D}(\cdot)$ to generate the output $\mathbf{y}$.

## B.2    SEPARATE FORCE-POSITION CONDITIONAL DIFFUSION MODEL:

In the first stage, we follow the previous force-control generator method by inputting the latent representation of the initial tactile image $\mathbf{z}^{(\mathbf{x})}$ and the target force change $\mathbf{\Delta f}$ into the force-control generator $\mathcal{F}_f(\cdot)$ to produce a latent representation of the tactile image $\mathbf{z}^{(\mathbf{x}')} = \mathcal{F}_f(\mathbf{z}^{(\mathbf{x})}, \mathbf{\Delta f})$ that satisfies the target force. In the second stage, this generated latent representation $\mathbf{z}^{(\mathbf{x}')}$, along with the latent representation of the contact mask $\mathbf{z}^{(\mathbf{c})}$, is input into the position-control generator $\mathcal{F}_p(\cdot)$. The output $\mathbf{z}^{(\mathbf{y})} = \mathcal{F}_p(\mathbf{z}^{(\mathbf{x}')}, \mathbf{z}^{(\mathbf{c})})$, which satisfies both the target force and contact position, is then decoded to produce the final tactile image $\mathbf{y} = \mathcal{D}(\mathbf{z}^{(\mathbf{y})})$.

# C    ROBUSTNESS VALIDATION OF CONTACT MASK ALIGNMENT

To evaluate the sensitivity of position control to inaccuracies in contact mask alignment, we conducted a series of controlled experiments by applying perturbations in three forms: scaling (S), translation (T), and rotation (R). The performance was measured using MSE and SSIM, and the results are summarized in Tables 4 and 5. For individual perturbations, scaling the mask within the range of 0.8 to 1.2 produced only negligible variations in both MSE and SSIM, suggesting that the method is largely insensitive to scale changes. Translation up to 4 pixels and rotation up to 2 degrees, which correspond to the maximum alignment errors in practice, also resulted in no significant degradation in the quality of the generated outputs. Even when scaling, translation, and rotation perturbations were combined, the generated results remained stable and acceptable. It should be emphasized that the contact mask is primarily used to determine the contact position, whereas the contact area is governed by force control and is therefore unaffected by such perturbations.

Table 4: Individual Perturbation Analysis with MSE↓ (lower is better) and SSIM↑ (higher is better)

| Perturbation Type | Parameter | MSE↓ | SSIM↑ |
|---|---|---|---|
| Scaling (S) | 1.0 | 23 | 0.83 |
| | 1.1 | 23 | 0.83 |
| | 1.2 | 24 | 0.83 |
| | 0.9 | 23 | 0.83 |
| | 0.8 | 24 | 0.83 |
| Translation (T) | 2 px | 23 | 0.83 |
| | 4 px | 24 | 0.83 |
| | 6 px | 25 | 0.82 |
| Rotation (R) | 2° | 23 | 0.83 |
| | 4° | 25 | 0.82 |
| | 6° | 27 | 0.82 |

Table 5: Combined Perturbation Analysis with MSE↓ (lower is better) and SSIM↑ (higher is better)

| Perturbation Combination | MSE↓ | SSIM↑ |
|---|---|---|
| S 0.9 + T4 + R2 | 23 | 0.82 |
| S 1.1 + T4 + R4 | 26 | 0.82 |
| S 0.8 + T6 + R6 | 28 | 0.82 |
| S 1.1 + T6 + R6 | 28 | 0.82 |

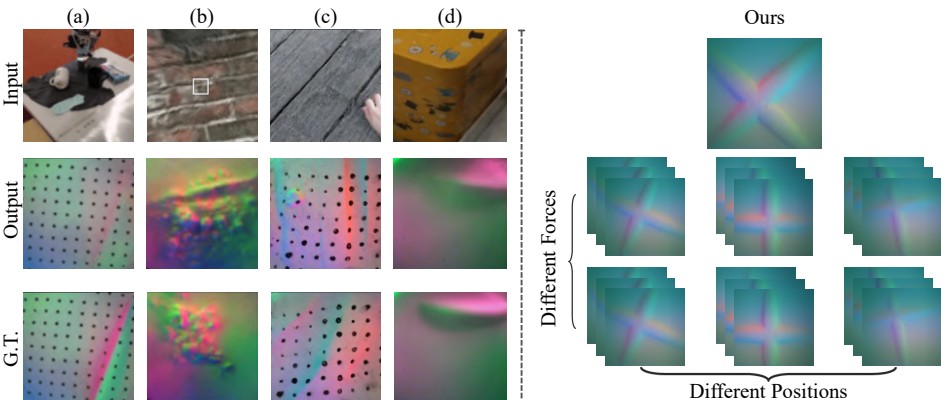

Figure 7: **Left** refer to different vis2tac generation pipeline, where (a) is VisGell (Li et al., 2019), (b) is TaRF (Dou et al., 2024), (c) is (Yang et al., 2023), and (d) is Touching a NeRF Zhong et al. (2022). **Right** refer to our method CONTROLTAC. Compared to other method, with a reference tactile image, CONTROLTAC can generate different images cover various positions and forces.

# D    ADDITIONAL EXPERIMENTS

## D.1    COMPARISON WITH VISION TO TACTILE GENERATION

To highlight the difference between CONTROLTAC and the previous Vis2Tac methods, we demonstrates more examples for different frameworks as shown in Fig. 7. Unlike vision to tactile generation pipeline, which only can generate a single unrealistic image from the reference contact position, CONTROLTAC can generate various realistic tactile images cover different contaction positions and forces.

## D.2    ADDITIONAL GENERATION QUALITY EVALUATION

Table 6: Effect of excluding the thin cylinder on model performance (MSE↓ lower is better, SSIM↑ higher is better)

| Dataset | MSE↓ | SSIM↑ |
|---|---|---|
| Complete Dataset | 23 | 0.83 |
| Excluding Thin Cylinder | 25 | 0.82 |

We conduct an experiment to investigate the impact of excluding the thin cylinder from the dataset. As shown in Table 6, removing the thin cylinder leads to a slight increase in MSE from 23 to 25 and a minor decrease in SSIM from 0.83 to 0.82, indicating that the overall performance remains roughly the same even without this object.

We further evaluate the model on unseen object generation to assess its generalization capabilities. Table 7 presents the results for generating tactile images of the YCB-object (Calli et al., 2015) nectarine and banana. Despite being unseen during training, the model achieves MSE and SSIM values comparable to those on the original dataset and also surpasses both the baseline models and the Taxim Si & Yuan (2022) simulator, demonstrating its ability to generate high-quality tactile images for previously unseen objects.

Table 7: Model performance on generating tactile images of unseen objects (MSE ↓ lower is better, SSIM ↑ higher is better)

| Object | Model | MSE | SSIM |
|--------|-------|-----|------|
| Nectarine | ControlTac | 25 | 0.80 |
| Nectarine | Hybrid | 35 | 0.75 |
| Nectarine | Separate | 190 | 0.72 |
| Nectarine | Taxim | 978 | 0.69 |
| Banana | ControlTac | 27 | 0.78 |
| Banana | Hybrid | 39 | 0.74 |
| Banana | Separate | 207 | 0.72 |
| Banana | Taxim | 1152 | 0.68 |

### D.3 IMPACT OF FORCE AUGMENTATIONS

To highlight the importance of force augmentations, we train a force estimator on different mixes of real and generated data. Specifically, we use 1,000 real contact positions and augment them with 20 or 40 force levels via the force-control generator, resullting in 20k and 40k synthetic samples. We then co-train estimators with varying amounts of real data (1k–20k) plus these augmented sets. As shown in Fig. 8, adding large amounts of generated data substantially reduces MAE when real data is scarce (e.g., only 1k samples), outperforming training on real data alone.

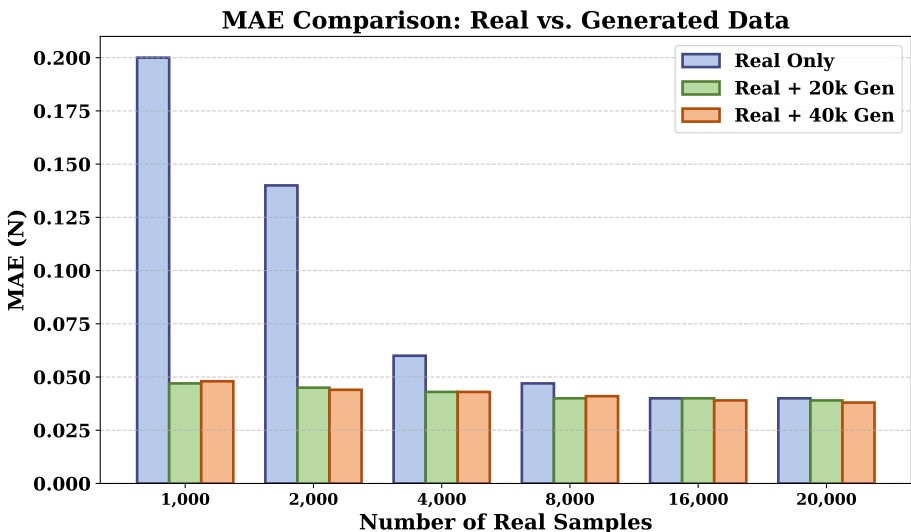

Figure 8: Force estimation performance (MAE) across different quantities of real and generated data. The normal force range is 1–10 N.

Moreover, with the generated dataset, the model achieves the same performance as training on the full real dataset (20,000 images) with only 8,000 real images. This suggests that the generated data effectively enrich the force distribution at each contact position, enhancing the training of the force estimator. Furthermore, combining larger quantities of both real and generated data yields the best performance, which highlights the realism and utility of the generated data.

### D.4 IMPACT OF DATA COMPOSITION ON MODEL PERFORMANCE

In our experiments, we observe that training the model with a combination of all-angle real data and generated data resulted in slightly worse performance compared to using only real data. This can be explained from two perspectives.

(1) First, the performance of the force estimator model is limited by its own capacity. As shown in Table 8, when we trained the model with varying amounts of real data (from 10k to 15k), we found that the MAE improvement plateaued once the data size exceeded 13k, indicating that adding more real data did not lead to significant gains. (2) Second, although the generated data is generally of high quality, it inevitably contains small errors. As the proportion of generated data increases, these errors tend to accumulate and negatively impact model training. Specifically, when training solely on different amounts of generated data, the MAE fluctuates as the data size increases, suggesting the presence of error accumulation.

Nevertheless, the errors in the generated data are minor and do not cause a significant drop in overall model performance. This indicates that while excessive generated data can "dilute" the contribution of real data, it does not fundamentally compromise the results (see Table 8). Furthermore, even when using a much larger amount of generated data (45k or 60k) in combination with real data, the performance does not deteriorate excessively, alleviating concerns about the quality of the generated data.

Table 8: MAE↓ (lower is better) of Force Estimator under Different Data Combinations. Gen refers to data generated by CONTROLTAC, while Real refers to force estimator trained by real data from FeelAnyForce (Shahidzadeh et al., 2025).

| Training Data Type | Data Size | MAE ↓ |
|---|---|---|
| Real | 10k | 0.061 |
| Real | 11k | 0.057 |
| Real | 12k | 0.055 |
| Real | 13k | 0.051 |
| Real | 14k | 0.054 |
| Real | 15k | 0.053 |
| Gen | 15k | 0.21 |
| Gen | 30k | 0.17 |
| Gen | 45k | 0.18 |
| Gen | 60k | 0.16 |
| Real + Gen | 15k + 15k | 0.055 |
| Real + Gen | 15k + 30k | 0.060 |
| Real + Gen | 15k + 45k | 0.058 |
| Real + Gen | 15k + 60k | 0.061 |

## D.5 COMPLETE COMPARISON OF POSE ESTIMATION PERFORMANCE

Fig. 9, Fig. 10 and Fig. 11 provide a comprehensive comparison of pose estimation errors under different training data settings, measured in pixels for X and Y coordinates and in degrees for orientation. The results are reported for four object categories: Cylinder (with three types), Cross, unseen T-shape, and the Type-C connector. Overall, pose estimators trained solely on generated data achieve strong performance and often outperform models trained on real data, even when the real dataset is relatively large. Simulation-based data from Taxim (Si & Yuan, 2022) consistently underperforms compared to both real and generated data, and combining real with simulation data generally offers limited improvements or even degrades performance due to the lower realism of simulated samples. Using varying force during data generation further improves performance over fixed-force generation since it can cover more variations in the real-world scenario. Notably, for the unseen T-shape object and Type-C connector, models trained on generated data generalize effectively, demonstrating the value of generated tactile images in covering diverse contact positions and angles.

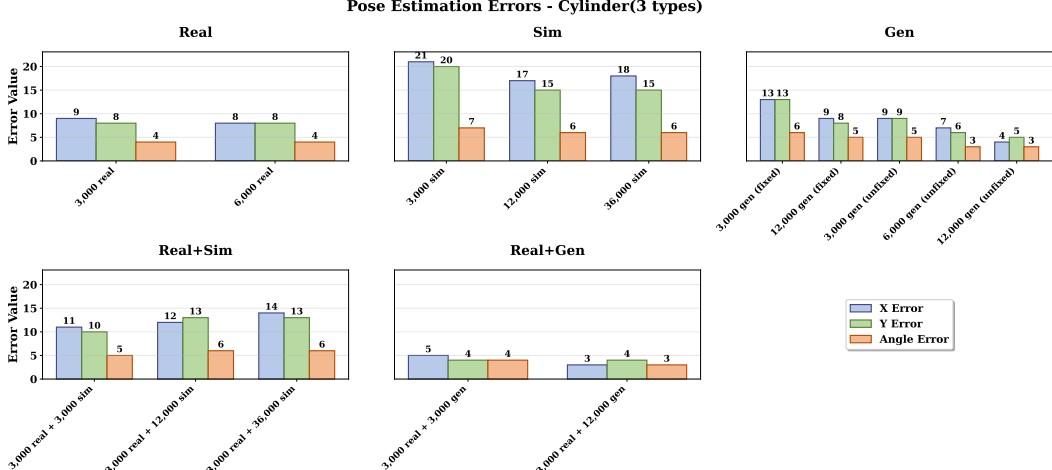

Figure 9: Pose estimation errors (in pixels and degrees) for cylinder objects under different training data regimes, grouped by real, simulated, real+sim, generated, and real+gen datasets. The y-axis is shared across groups to enable scaling comparison.

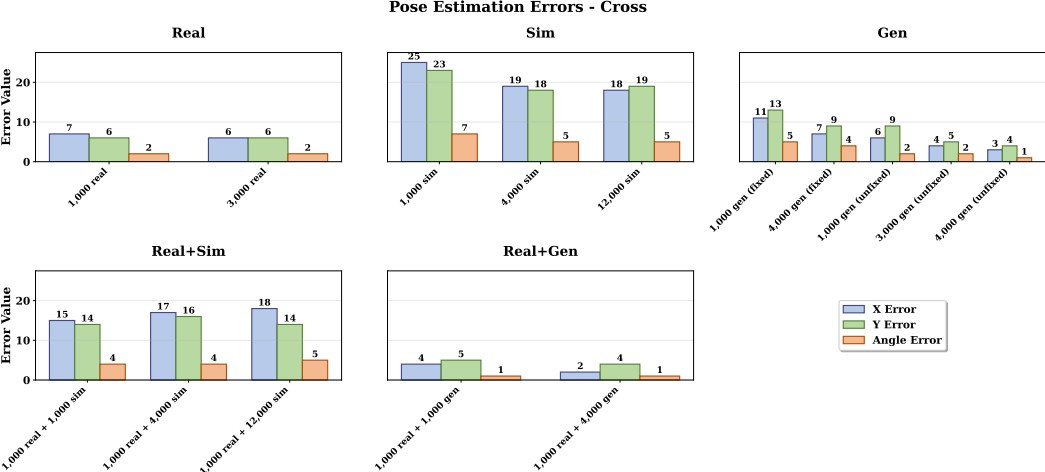

Figure 10: Pose estimation errors (in pixels and degrees) for the cross object under different training data regimes, grouped by real, simulated, real+sim, generated, and real+gen datasets. The y-axis is shared across groups to enable scaling comparison.

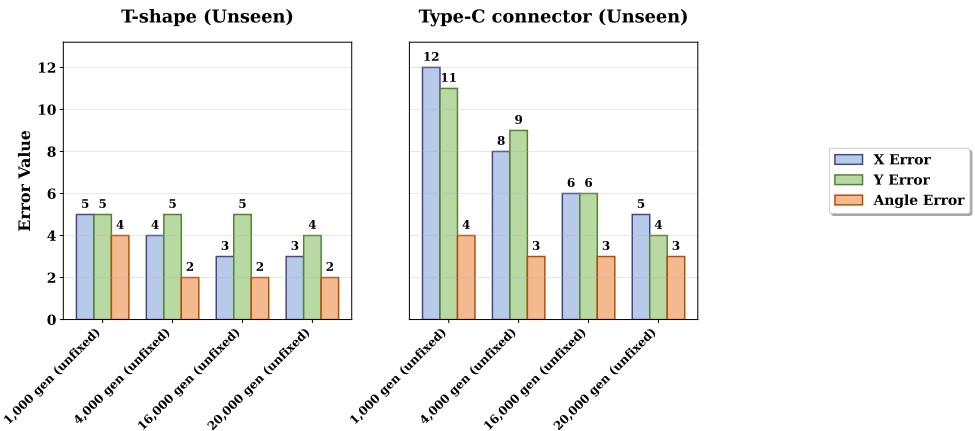

Figure 11: Pose estimation errors (in pixels and degrees) for the unseen T-shape and Type-C connector under different training data regimes, grouped by real, simulated, real+sim, generated, and real+gen datasets. The y-axis is shared across groups to enable scaling comparison.

## D.6 DETAILS OF OBJECT PUSHING

Table 9 summarizes the evaluation of the force estimator on the object-pushing task. The objective of this downstream task is to evaluate whether a force estimator trained on ControlTac-generated data can achieve performance comparable to one trained on real data, thereby highlighting the high quality of the data generated by ControlTac. The table provides two sets of numbers for each object; for instance, in the entry "Bottle (0.63)", 0.63 refers to the mass of the glass bottle in kilograms. The other numbers indicate the measured and predicted pushing forces: $1.08\,\mathrm{N}$ is the force measured by the force sensor, $1.14\,\mathrm{N}$ is the force predicted by the estimator trained on real-world data, and $1.16\,\mathrm{N}$ is the force predicted by the estimator trained on ControlTac-generated tactile data. During the pushing process, the applied force is dynamic: as the robotic arm pushes the object at a constant speed, force sensor readings and corresponding tactile data are recorded and used by the estimator to predict the pushing force. Across all four objects—including the metal weight, water-filled cylinder, and glass bottle—the estimator trained on generated images achieves performance comparable to that of the real-data-trained model. This demonstrates that training with generated data enables the force estimator to generalize effectively to diverse objects with varying textures, materials, and weights, closely matching the accuracy of the real-data-trained model.

Table 9: Results of object pushing experiments for the four objects.

| Force [N] | Weight (1.0) | Cyl. (0.50) | Cyl. (0.56) | Bottle (0.63) |
|---|---|---|---|---|
| Force ATI (G.T.) | 2.24 | 0.96 | 1.06 | 1.08 |
| Force (Real Data) | 2.38 | 1.08 | 1.18 | 1.14 |
| Force (Ours) | 2.36 | 1.11 | 1.17 | 1.16 |

## D.7 DETAILS OF INSERTION TASK

For the precise insertion task, we 3D print three different objects and a hole: a (7 cm-long) cylinder with a diameter of (7 mm), a (7 cm) by (3 cm) cross-shaped object with (7 mm) diameter, a (7 cm) by (3 cm) T shape object with (7 mm) diameter, and a hole measuring (5 cm) in height and (3 cm) in depth with (10 mm) diameter. For the USB insertion task, we let the robot to insert the type-c cable into a charger. To finish the insertion task, we let the XArm7 with two Gelsight Mini grasp the object above the hole with a random angle and in-hand position and then adjust the pose and position according to the estimation result. The setting is shown in Fig. 6.

For the task setting, the hole has been set up in a known position, where the robot can reach the location above it. To finish the insertion, the robot need to adjust it's in-hand pose according to its initial grasping. Specifically, the pose estimator first predict the object's pose on the tactile sensor. Then, we compute the Euclidean distance from the estimated pose to the center. This distance is converted from pixel units to real-world units using a scaling factor of 1 pixel = $\frac{1}{20}mm$. For estimation-based robotic control, the robot adjusts its end-effector by rotating along the Rx axis and translating along the y-axis based on the predicted pose, which align with the object vertically above the hole.

For the insertion task, which is a real-robot application of the pose estimation task, we can evaluate errors in both position and orientation. If the generated data is of insufficient quality, the estimator's performance will degrade, making it unsuitable for the insertion task. Therefore, monitoring the performance of the pose estimator provides an effective way to mitigate potential safety issues during insertion.

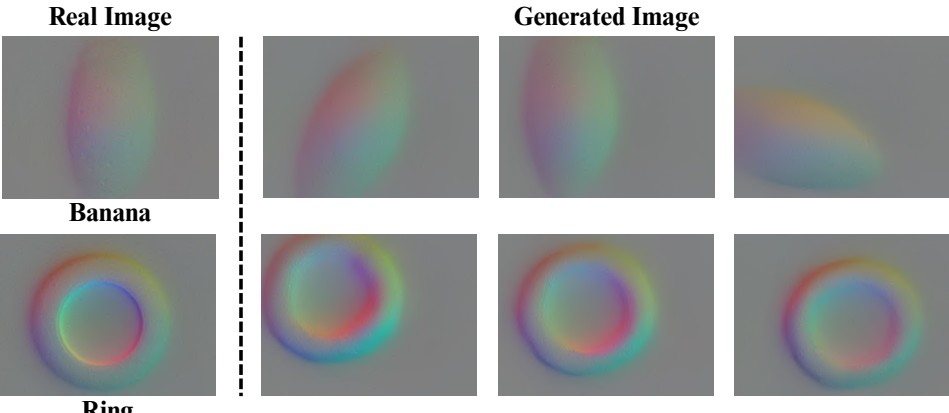

Figure 12: Failure cases on banana and flattened ring.

## E  FAILURE ANALYSIS

We acknowledge certain limitations arising from the restricted diversity of the training set, where all contact objects are made of PLA and predominantly exhibit curved surfaces. Consequently, the model shows weaker generation performance for objects with flat surfaces, rich textures, or varying hardness, such as flattened rings and bananas, as illustrated in Fig. 12. For clearer visualization, we subtract the background and apply a constant offset of 127 to shift pixel values into a valid display range. To address these limitations, we augmented the training set of 20,000 samples with 1,000 additional tactile images of flat-surfaced cubes. This targeted addition led to a clear improvement in generation quality for previously unseen flattened rings (MSE reduced from 35 to 27; SSIM increased from 0.80 to 0.83), demonstrating that even a relatively small amount of domain-specific data can substantially enhance performance in underrepresented scenarios.

## F  CLASSIFIER ARCHITECTURES

### F.1  CNN CLASSIFIER

We design a convolutional neural networ (CNN) for image classification, consisting of four convolutional blocks followed by two fully connected layers. The architecture is as follows:

- **Input**: RGB images of shape $(3, 224, 224)$
- **Convolutional Block 1**:
  - Conv2d: $3 \rightarrow 32$, kernel size $3 \times 3$, stride 1, padding 1
  - BatchNorm2d

- – ReLU activation
- – MaxPool2d: $2 \times 2$
- **Convolutional Block 2**:
  - – Conv2d: $32 \rightarrow 64$
  - – BatchNorm2d
  - – ReLU activation
  - – MaxPool2d: $2 \times 2$
- **Convolutional Block 3**:
  - – Conv2d: $64 \rightarrow 128$
  - – BatchNorm2d
  - – ReLU activation
  - – MaxPool2d: $2 \times 2$
- **Convolutional Block 4**:
  - – Conv2d: $128 \rightarrow 256$
  - – BatchNorm2d
  - – ReLU activation
  - – MaxPool2d: $2 \times 2$
- **Flatten Layer**: Feature map of shape $(256, 14, 14)$ is flattened to $(50176)$
- **Fully Connected Layers**:
  - – Linear: $50176 \rightarrow 512$
  - – ReLU + Dropout ($p = 0.5$)
  - – Linear: $512 \rightarrow 6$ (number of classes)

### F.2 ViT Classifier

We use the Vision Transformer (ViT) architecture (Dosovitskiy et al., 2020), specifically the `vit_base_patch16_224` variant implemented via the `timm` library (Wightman, 2019). This transformer-based model operates on image patches and employs self-attention mechanisms.

- **Patch Size**: $16 \times 16$
- **Input Resolution**: $224 \times 224$
- **Number of Patches**: 196 (i.e., $14 \times 14$ patches)
- **Transformer Encoder**:
  - – Embedding dimension: 768
  - – Number of transformer layers (depth): 12
  - – Number of attention heads: 12
  - – MLP dimension: 3072
- **Classification Head**: The original head is replaced with:
  - – Linear: $768 \rightarrow 6$
- **Pretraining Settings**:
  - – *ViT with Pretraining*: The model is initialized with weights pretrained on ImageNet 2012 (Deng et al., 2009), providing a strong starting point for transfer learning.
  - – *ViT without Pretraining*: The model is trained from scratch using random initialization, without access to any external datasets.

## G Addition Visualizations

In this section, we provide additional visualizations to clarify the concepts discussed.

## G.1 SENSITIVITY OF TACTILE IMAGES TO FORCE VARIATIONS

This figure illustrates the high sensitivity of the force estimation task to tactile image accuracy. We selected four force values—3.3 N, 3.8 N, 4.3 N, and 4.8 N—to show how subtle force differences affect tactile images. Differences smaller than 0.5 N are almost imperceptible, and even differences of 1 N result in only very subtle changes. This highlights the difficulty of accurately labeling tactile images with corresponding forces and underscores the challenge of using generated or simulated tactile images for this task.

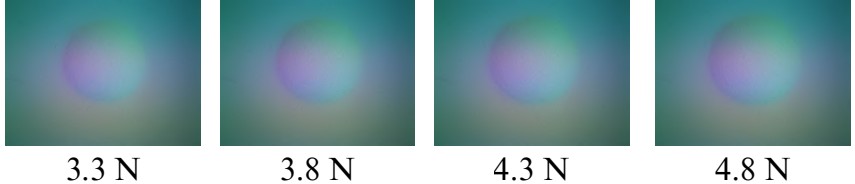

| 3.3 N | 3.8 N | 4.3 N | 4.8 N |

Figure 13: Tactile images corresponding to four different force values: 3.3 N, 3.8 N, 4.3 N, and 4.8 N.

## G.2 VISUALIZATION OF A SPECIAL CASE REQUIRING GENERATION

In this section, Fig. 14 visualize a simple special case, where the cylinder is placed in the corner. In this case, the sensor is only able to contact with the cylinder with the edge of the sensor, where it cannot cover the other contact positions. Notably, it's simple conditions, whereas it easily to happen in the real world with some articulate objects like a cabinet or a action figure.

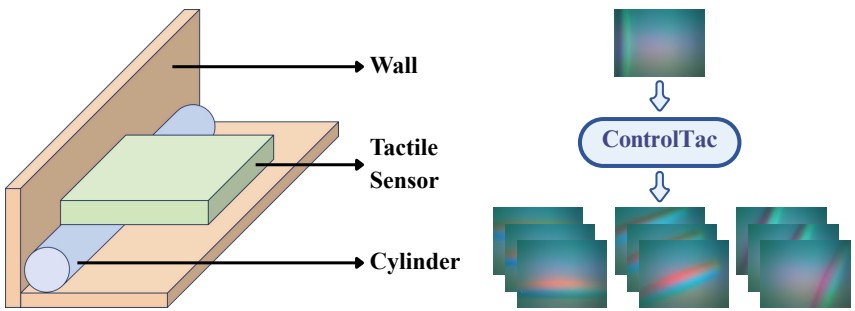

Figure 14: With the cylinder placed in the corner, the sensor only can get several images contact with the edge. In this case, CONTROLTAC can generate various images with more contact positions.

## G.3 VISUALIZATION OF VARIOUS POSITIONS OF DIFFERENT OBJECTS

In this section, we visualize the various positions of different objects. Fig. 15 provides valuable insights into how positions can lead to dramatic changes in the tactile image's color distribution.

## G.4 VISUALIZATION OF ERROR MAP

In this section, Fig. 16 illustrates the error map of CONTROLTAC compared to two baseline models. It is evident that CONTROLTAC significantly outperforms the other two baseline models.

## G.5 VISUALIZATION OF DEPTH MAP

In this section, Fig. 17 and Fig. 18 show the visualizations of the depth derived from tactile images generated by CONTROLTAC, along with the corresponding ground truth and error maps, to verify that the generator does not merely scale overall brightness.

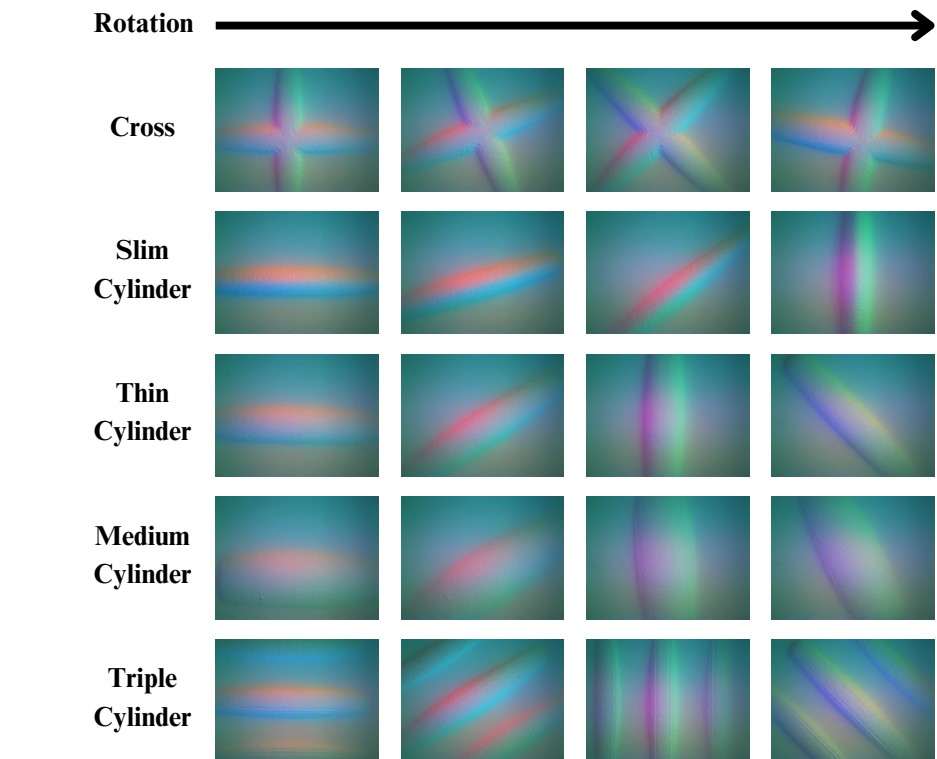

Figure 15: **Visualization of various positions.** Note: The rotational symmetry of spheres renders their angular representations redundant, and thus they are not included here.

### G.6 Visualization of Generated Image using Force-Control Generation Component

In this section, we showcase the visualization results using the force-control generation component of CONTROLTAC. Fig. 19 presents the generated tactile images for the same contact position, demonstrating excellent results and the effectiveness of this component.

### G.7 Visualization of different sensor samples

In this section, we visualize the difference of contact image between different sensor samples, as shown in Fig. 20.

### G.8 Analysis of Simulated Tactile Images

In this section, we present tactile images generated using Taxim (Si & Yuan, 2022). As shown in Fig. 21, the simulated images lack realism, highlighting the limitations of current simulation methods for tactile data.

### G.9 Object and Tactile Image Visualization for Classification

In this section, we present six objects used in the classification task along with their corresponding tactile images, as shown in Fig. 22.

### G.10 Visualization of the Type-C Connector Insertion Task

To better illustrate the Type-C connector insertion task, Fig. 23 shows both the real Type-C connector and its corresponding tactile image used in this experiment.

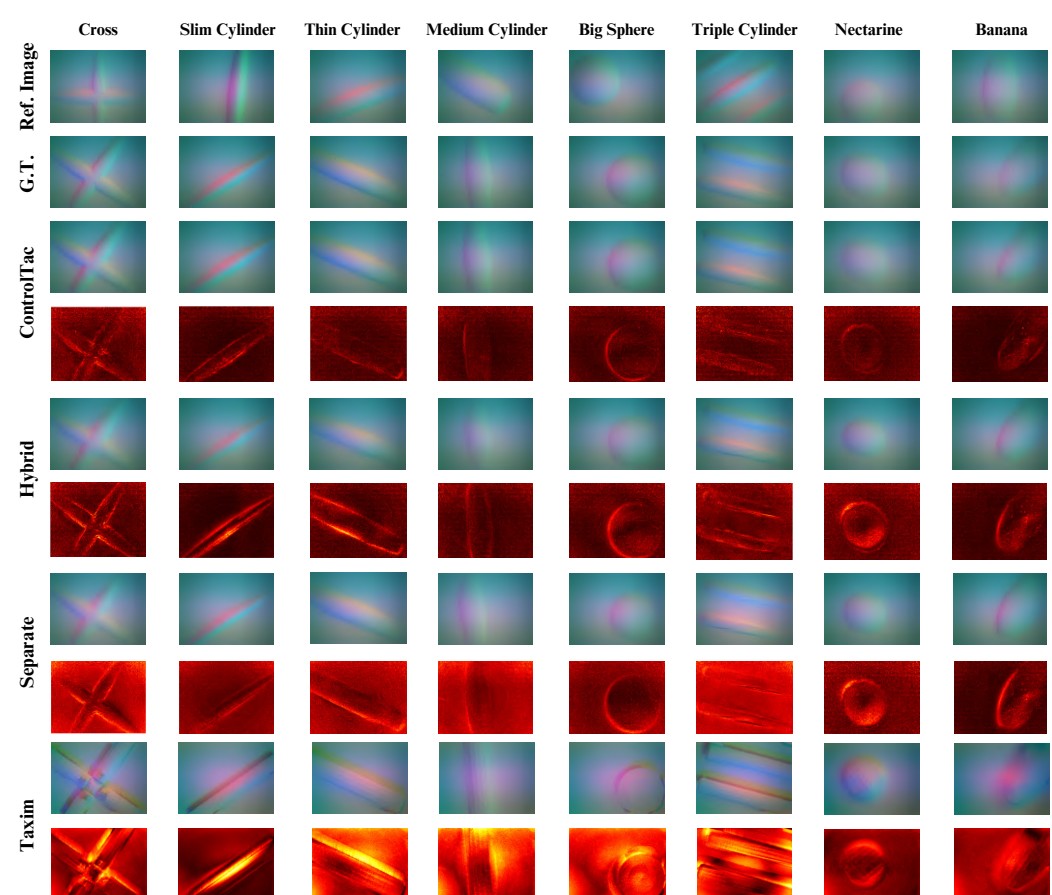

Figure 16: Error map comparison between CONTROLTAC and two baseline models.

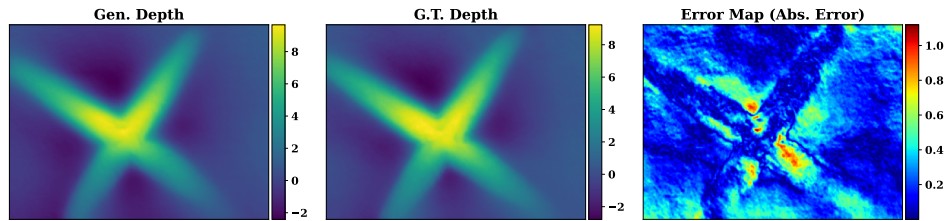

Figure 17: Depth maps of the Cross object generated from CONTROLTAC-generated tactile images (Gen.), compared with Ground Truth (G.T.) and corresponding error maps.

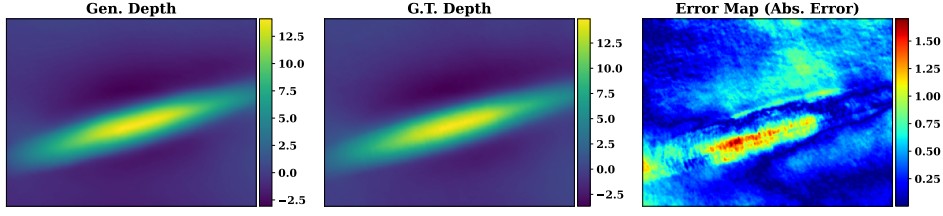

Figure 18: Depth maps of the Thin Cylinder object generated from CONTROLTAC-generated tactile images (Gen.), compared with Ground Truth (G.T.) and corresponding error maps.

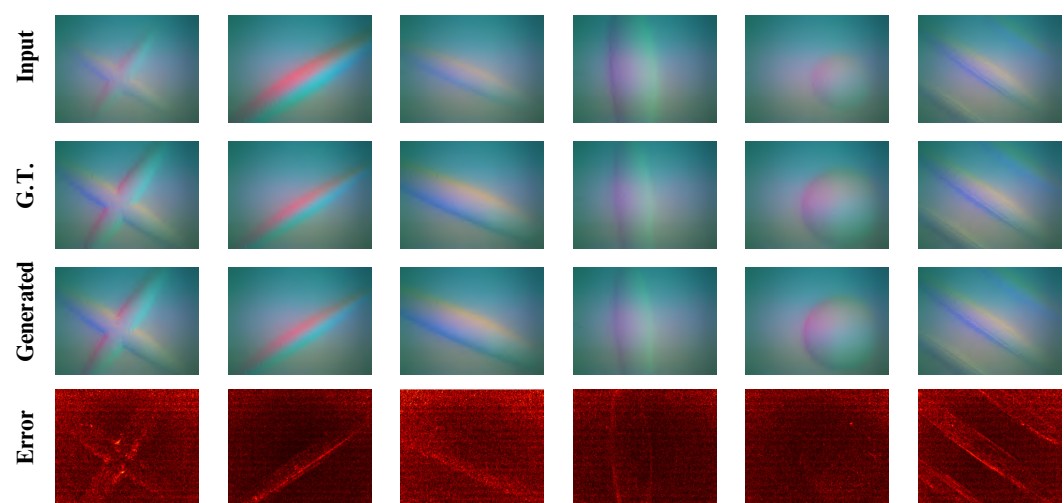

Figure 19: Generated tactile images using the force-control generation component of CONTROLTAC at the same contact position.

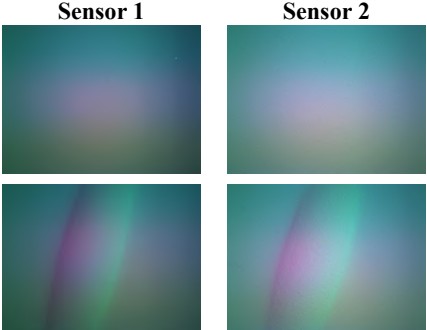

Figure 20: The background of Sensor 1 is noticeably darker than that of Sensor 2.

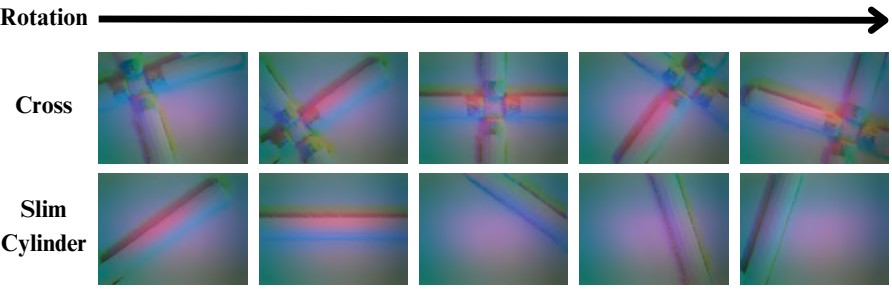

Figure 21: Simulated tactile images using Taxim (Si & Yuan, 2022).

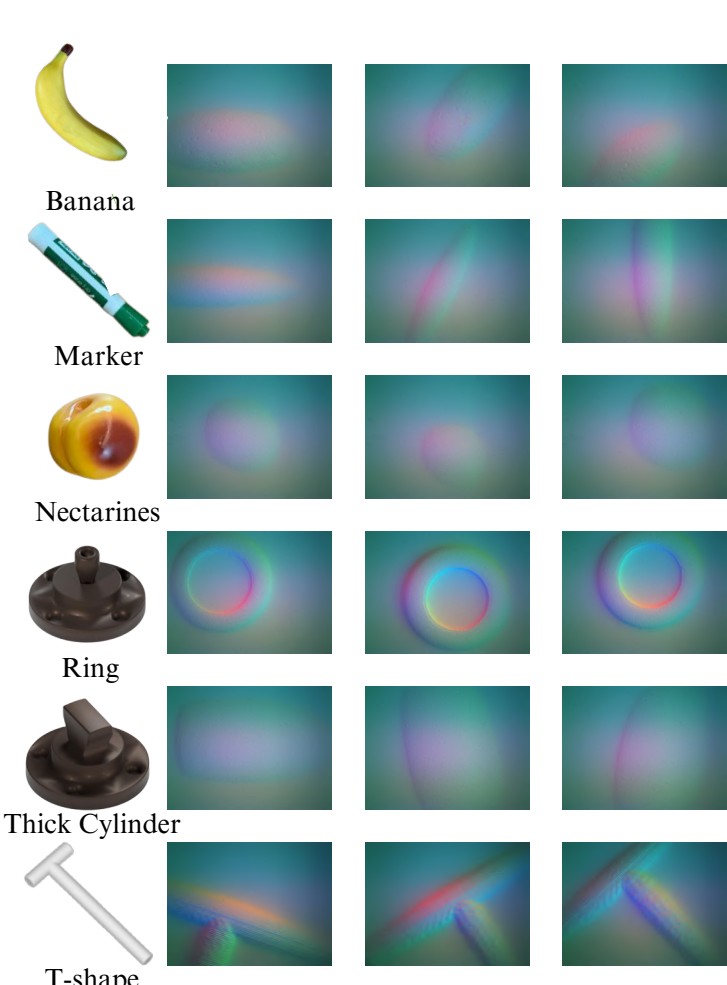

Figure 22: Six objects and their corresponding tactile images used in the classification task.

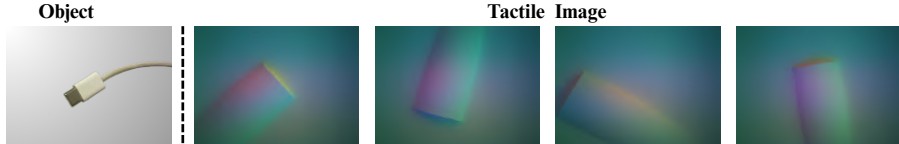

Figure 23: Real USB Type-C connector and corresponding tactile image.

