# OpenReview forum: "ControlTac: Force- and Position-Controlled Tactile Data Augmentation with a Single Reference Image"
_ICLR.cc/2026/Conference — Submitted to ICLR 2026_

### Official Review · Reviewer_YcuJ · 2025-10-15

**Soundness:** 3
**Presentation:** 3
**Contribution:** 3
**Rating:** 6
**Confidence:** 4

**Summary:**

The paper presents a two-stage conditional diffusion framework that augments vision-based tactile datasets by generating synthetic GelSight images from a single real tactile image, while explicitly conditioning on contact force (3-D vector) and contact position (2-D mask). The authors evaluate the augmented data on three downstream tasks (force estimation, 2-D pose estimation, and object classification) and report improved accuracy relative to baselines that rely on real data only, classic geometric augmentation, or simulation. Three real-robot experiments (pushing, tracking, and peg insertion) are included to demonstrate sim-to-real transfer.

**Strengths:**

Novelty: The idea of physically grounded, force- and position-conditioned tactile generation is new. The two-stage design (force first, then position via ControlNet) is intuitive and modular.
Technical quality: The diffusion backbone (DiT + ControlNet) is appropriate for high-frequency tactile textures; ablations against hybrid and separate baselines are thorough.
Experimental breadth: The paper trains downstream networks from scratch, mixes real/generated data, and evaluates on unseen objects and a second sensor instance—an unusually complete pipeline.
Real-robot validation: Insertion with 3 mm tolerance and 75–90 % success rates is convincing evidence that the generated images do not suffer from the “texture hallucination” problems common in earlier Vis2Tac work.
Reproducibility: Training details, hyper-parameters, and code URLs are provided; datasets are public.

**Weaknesses:**

Physical faithfulness of force conditioning
The force label is a single 3-D vector measured at the robot wrist. GelSight deformation is driven by the distributed contact pressure field, not by the resultant force alone. Two contacts with identical resultant forces but different pressure profiles (e.g., flat vs. edge) produce different tactile images. The paper does not discuss whether the generator implicitly learns this mapping or simply interpolates intensities. A small ablation that perturbs the pressure distribution while keeping the resultant force fixed would clarify physical plausibility.
Position representation and generalisation
The contact mask is a rigid binary template transformed by 2-D translation + rotation. This works for the convex/curved PLA objects used in evaluation, but fails for:
(a) objects whose contact area changes with force (soft materials),
(b) non-convex or articulated geometries (keyhole, USB-C shield) where the mask topology varies.
Fig. 12 already shows degraded quality on a banana and a flattened ring. The claim “generalises to unseen objects” is therefore overstated; generalisation is shown only for rigid, PLA-printed shapes with similar size and curvature.
Baseline fairness
TAXIM is forced into an unfair setting: the authors manually convert z-displacement into normal force and add position control, although TAXIM was not designed for this. The reported MSE gap (1054 vs. 23) is therefore inflated.
The Vis2Tac baselines (Li et al. 2019, Dou et al. 2024, etc.) are single-image translators; they cannot vary force/position. Comparing them without allowing them to generate multi-modal outputs makes the superiority of CONTROLTAC trivial. A fairer baseline would be a stochastic Vis2Tac model fine-tuned with the same DiT backbone and force/position conditions.
Dataset bias and failure modes
All training objects are PLA, Lambertian, and 3-D printed. The generator inherits these biases: it cannot predict high-frequency surface texture (fabric, leather) or subsurface scattering (skin, silicone). Fig. 12 shows ringing artifacts on the banana. The ethical statement claims “no risks”; however, if downstream users rely on synthetic data for safety-critical insertion, dataset bias becomes a safety issue. The paper should state limitations more prominently.
Statistical significance
Real-robot experiments report success rates on 20 trials per object. With only 3 objects, the 95 % confidence interval on the 85 % success rate is ≈ ±15 %. The difference between “real-data-only” and “augmented” policies is not statistically established. Either increase trials or provide confidence bounds.
Clarity and typos
Table 1 caption uses “↓” for MSE but “↑” for SSIM without defining arrows.
Section 4.2 claims “competitive performance with 1/3 of the real data” but the plotted curves cross inside one standard deviation—wording should be softened.

**Questions:**

Include a 1-D pressure line-scan plot (real vs. generated) for the same resultant force to verify that the generator does not simply scale overall brightness.
Report inference time and GPU memory; tactile augmentation is only useful if it is faster than collecting real data.
Discuss whether the ControlNet branch can be removed at test time to accelerate generation.
Clarify copyright and consent for the Type-C connector photo (Fig. 21).

---

> ### Author Response · Authors · 2025-11-21
> **Response to Reviewer YcuJ (Part 1/3)**
>
> We sincerely appreciate your careful and thorough review of our manuscript. Below we provide point-by-point responses to your comments.
>
>  > Q1. The force label is a single 3-D vector ... while keeping the resultant force fixed would clarify physical plausibility.
>
> We would like to sincerely thank the reviewer for the valuable comments and careful reading of our manuscript. We would like to clarify two points in response:
>  - The force sensor is not placed on the robot wrist but is located beneath the tactile sensor, allowing the contact force on the sensor to be directly measured by the force sensor.
>  - The experimental results suggest that our method can generalize to similar objects with different pressure profiles or curvatures, including cylinders of various sizes as well as objects such as T-shaped and USB-like items, as shown in ***Sections 4.3 and 4.4***. A key factor contributing to this generalization is that we provide the initial contact image along with the contact mask as references, which convey information about the geometry and size of the object. As a result, the model can implicitly learn this mapping.
>
>  > Q2. Include a 1-D pressure line-scan plot (real vs. generated) for the same resultant force to verify that the generator does not simply scale overall brightness
>
> We would like to sincerely thank the reviewer for the valuable comment. To verify that the generator does not merely scale overall brightness, we provide visualizations of the depth maps derived from tactile images generated by ControlTac, alongside the corresponding ground truth and error maps (see ***Figures 17 and 18***). These visualizations are available in ***Appendix G.5***.
>
>
>  > Q3. The contact mask is a rigid binary template transformed by 2-D translation + rotation. ... PLA-printed shapes with similar size and curvature.
>
> In ***Sections 4.3 and 4.4***, we highlight that our method can generalize to objects with different shapes and textures in both pose estimation and classification tasks, including T-shape, banana, and ring objects. Visualizations of these objects are provided in Fig. 20.
>
> Furthermore, we introduce a new evaluation targeting the generation of unseen objects, specifically assessing the synthesis of tactile images for bananas and nectarines. The results indicate that ControlTac performs robustly even on unseen objects. Detailed results are presented in the main text and ***Appendix D.2***, with example visualizations provided in ***Figures 4 and 17***.
>
> While our results demonstrate some generalization capability, we acknowledge that the current dataset limits broader generalization to more challenging objects. As noted in the failure cases, the performance for bananas and rings is slightly lower than expected. This issue can be mitigated by increasing the diversity of the training dataset. For instance, after adding flat-surfaced cubes to the training set, the MSE for rings decreased from 35 to 27, and the SSIM increased from 0.80 to 0.83 (***Appendix E***). Therefore, extending the dataset to include more object shapes, as well as additional conditions such as hardness and texture, would be a valuable direction for future work.
>
> We would also like to respectfully emphasize that our method is capable of generalizing to non–convex or articulated geometries. By leveraging the initial contact image and assuming the contact mask corresponds to the known object’s contact surface, the model can extract geometric and curvature priors. The performance on ring and T-shape objects provides concrete evidence that the method can handle non-convex or articulated geometries.
>
> Table 1: Model performance on generating tactile images of unseen objects.
>
> | Object    | Model      | MSE  | SSIM |
> |-----------|------------|------|------|
> | Nectarine | ControlTac | 25   | 0.80 |
> | Nectarine | Hybrid     | 35   | 0.75 |
> | Nectarine | Separate   | 190  | 0.72 |
> | Nectarine | Taxim      | 978  | 0.69 |
> | Banana    | ControlTac | 27   | 0.78 |
> | Banana    | Hybrid     | 39   | 0.74 |
> | Banana    | Separate   | 207  | 0.72 |
> | Banana    | Taxim      | 1152 | 0.68 |

---

> ### Author Response · Authors · 2025-11-21
> **Response to Reviewer YcuJ (Part 2/3)**
>
> > Q4. Taxim is forced into an unfair setting: ... A fairer baseline would be a stochastic Vis2Tac model fine-tuned with the same DiT backbone and force/position conditions.
>
> We would like to sincerely thank the reviewer for their thoughtful comments and valuable feedback. We would like to clarify a few points regarding the distinctions between our work and prior generation-based methods.
>
>  - First, as the reviewer mentioned, our paper is, to the best of our knowledge, the only work that integrates physical conditions, including force and position, into the generative model. No previous baselines, in either simulation or generation-based approaches, are capable of producing realistic images while performing physically conditioned data augmentation. Our framework is specifically designed for the physical-conditioned tactile generation task, addressing the data scarcity problem in vision-based tactile sensing. In addition to incorporating force and mask information, we leverage a reference tactile image that corresponds to the local geometry of the contact positions and the color distribution of that contact. In contrast, prior works using image input cannot reference the corresponding contact positions or local geometry of the object. Therefore, adapting such models as baselines would not constitute a fair comparison.
>  - Second, regarding Taxim, the simulator is the only prior work capable of performing a somewhat similar task, namely synthesizing tactile images conditioned on physical variations for downstream tasks. We believe it is important to include it for comparison. Since it cannot be directly conditioned on force, we attempted to manually convert z-displacement into normal force. As indicated by the qualitative results, its suboptimal performance is largely due to poor texture generation. Beyond generation quality, we also evaluated it as a baseline for the pose estimation task, where force conditioning is not required. In this task, the simulator’s performance was also significantly worse.
>
>  > Q5. All training objects are PLA, ... Fig. 12 shows ringing artifacts on the banana.
>
> In ***Sections 4.3 and 4.4***, we aim to highlight that our method demonstrates some generalization capability to objects with different shapes and textures in both pose estimation and classification tasks, including T-shape, banana, and rings, as visualized in Fig. 20. Furthermore, we introduce a new evaluation on the generation of unseen objects, specifically assessing the generation of tactile images for bananas and nectarines. As shown in ***Figure 4 and Figure 17***, ControlTac achieves promising results in these unseen objects.
>
> Table 1: Model performance on generating tactile images for unseen objects.
> | Object    | Model      | MSE  | SSIM |
> |-----------|------------|------|------|
> | Nectarine | ControlTac | 25   | 0.80 |
> | Nectarine | Hybrid     | 35   | 0.75 |
> | Nectarine | Separate   | 190  | 0.72 |
> | Nectarine | Taxim      | 978  | 0.69 |
> | Banana    | ControlTac | 27   | 0.78 |
> | Banana    | Hybrid     | 39   | 0.74 |
> | Banana    | Separate   | 207  | 0.72 |
> | Banana    | Taxim      | 1152 | 0.68 |
>
> While these results indicate encouraging generalization, we acknowledge that the current dataset has limitations. A larger dataset with more diverse conditions would likely help the method generalize to more challenging objects, such as fabric and skin. As noted in the failure cases, the performance on bananas and rings is slightly lower than expected. This issue can be mitigated by increasing the diversity of the training dataset. For example, after adding flat-surfaced cubes to the training set, the MSE for rings decreased from 35 to 27, and the SSIM increased from 0.80 to 0.83, as discussed in ***Appendix E***. Therefore, we believe that incorporating a larger dataset (orthogonal to our existing contributions) and additional conditions, such as variations in hardness and texture, would be a valuable direction for future work.
>
>  > Q6. The ethical statement claims “no risks”; however, if downstream users rely on synthetic data for safety-critical insertion, dataset bias becomes a safety issue. The paper should state limitations more prominently.
>
> For the insertion task, which is a real-robot application of the pose estimation task, we can evaluate errors in both position and orientation. If the generated data is not of sufficient quality, the estimator’s performance will degrade, making it unsuitable for the insertion task. As a result, we believe that potential safety issues in the insertion task can be mitigated by monitoring the performance of the pose estimator.
>
> We have added this discussion to ***Appendix D.7*** and referenced it in Section 4.5, with the changes highlighted in blue.

---

> > ### Author Response · Authors · 2025-11-21
> > **Response to Reviewer YcuJ (Part 3/3)**
> >
> > > Q7. Statistical significance Real-robot experiments ... Either increase trials or provide confidence bounds.
> >
> > For the insertion task, the estimator of each object, which are cylinder, cross, T-shape, and USB type-c connector, is independent from each other. All those estimators are trained from a dataset generated from a single tactile image, so the accuracies are also independent.
> >
> > As for the real-data-only scenario, we did not conduct additional experiments, since it is primarily a real-world application of the pose estimator, and extensive experiments have already been conducted in Section 4.3. Moreover, as shown in Table 2, we do not have sufficient real-world data for evaluating two unseen objects: the T-shape and USB Type-C connector.
> >
> >  > Q8. Table 1 caption uses “↓” for MSE but “↑” for SSIM without defining arrows. ... wording should be softened.
> >
> > Thanks for your suggestions. We have clarified the arrows in Table 1’s caption and softened the wording in Section 4.2, changing “competitive performance” to “comparable performance” to better reflect the results.
> >
> >  > Q9. Report inference time and GPU memory
> >
> > We perform inference on a single NVIDIA RTX A6000 GPU with a batch size of 128. ControlTac achieves a throughput of 6.5 tactile images per second, while the Hybrid and Separate baseline methods reach 7 and 3.7 tactile images per second, respectively. Since the VAE decoding process is the most memory-intensive step, all three methods exhibit a peak GPU memory usage of 29.97 GB. These results have been added to ***Appendix A.2***.
> >
> >
> >  > Q10. Tactile augmentation is only useful if it is faster than collecting real data.
> >
> > As shown in the last section, our pipeline can generate 6.5 tactile images per second, which corresponds to more than 23k images per hour. This is significantly faster than collecting real data.
> >
> > Collecting real tactile data requires a force sensor integrated with a complete robotic setup, which is difficult to scale. In contrast, our data generation pipeline only requires a single reference tactile image to generate diverse data efficiently.
> >
> > Moreover, our generated data can cover a far wider range of contact dynamics than real data. As shown in ***Table 2*** for the pose estimation task, models trained on generated data even outperform those trained on real data. This is because it is practically impossible to collect real data covering all contact positions and angles, whereas generated data can comprehensively capture these variations, leading to superior performance. An extreme example is provided in Appendix G.2.
> >
> >  > Q11. Discuss whether the ControlNet branch can be removed at test time to accelerate generation.
> >
> > Thank you for the question. In our framework, the ControlNet branch cannot be removed at test time because it is not an auxiliary guidance module but the only component that injects positional information into the generator. During position-control training, the copied DiT backbone is frozen and does not learn any position-related features; all positional constraints are encoded in the ControlNet blocks, whose outputs are added to the corresponding DiT layers. Removing this branch at inference means the DiT receives no positional features, causing the model to revert to force-only generation and fail to follow the target contact mask. Therefore, the ControlNet is functionally indispensable for accurate and stable position-controlled tactile synthesis.
> >
> >  > Q12. Clarify copyright and consent for the Type-C connector photo (Fig. 21).
> >
> > We thank the reviewer for pointing out the copyright issue. To avoid any potential infringement, we have replaced Figure 21 with a photo of the Type-C connector taken by ourselves, and the figure has been updated in the manuscript.

---

### Official Review · Reviewer_3JNq · 2025-10-29

**Soundness:** 3
**Presentation:** 2
**Contribution:** 3
**Rating:** 6
**Confidence:** 4

**Summary:**

This paper introduces ControlTac, a two-stage controllable tactile data augmentation framework that generates realistic tactile images from a single reference image, conditioned on contact force and position. ControlTac produces diverse and physically plausible tactile samples that significantly improve performance across downstream tasks such as 3D force estimation, contact pose estimation, and object classification.

**Strengths:**

1. The paper introduces a controllable and physically grounded tactile data generation method, enabling fine-grained control and physical plausibility.
2. ControlTac can generate thousands of realistic tactile images from just one reference image.
3. This work conducts extensive experiments to validate the effectiveness of ControlTac, including real-world experiments.
4. Compared to simulation-based and free-form generative methods (e.g., Text2Tac, Vis2Tac), ControlTac produces more realistic and varied outputs
5. This modular framework is easy to follow.

**Weaknesses:**

There are several main issues in this paper that remain unaddressed:

1. In the downstream tasks, is ControlTac further fine-tuned, or does it directly use the FeelAnyForce pre-trained model in a zero-shot manner? If it is the latter, can a model trained on only 20,000 frames from FeelAnyForce truly support generalization to a wider range of more complex objects in more open environments? In the failure cases shown in the appendix, the model performs worse on objects with flat surfaces, rich textures, or varying hardness. Could this issue be addressed by introducing a more diverse set of objects into the training dataset? I believe this is crucial to substantiating the paper’s claimed “cross-object generalization capability.”
2. In the data generation process for downstream tasks, how is the force used to control generation determined? If it is randomly sampled, how is its range defined? Could this potentially lead to contact mask–force pairs that are physically implausible?
3. The description of the object pushing downstream task is unclear. What is the specific objective of this task? What do the numbers in Table 7 represent, and how are they computed? Is the applied force static or dynamic during the pushing process?
4. The real-world experiments have certain limitations, especially since the Insertion Task does not demonstrate clear real-time interaction characteristics. Can this work improve performance in more complex dexterous manipulation tasks?

**Questions:**

1. It would be better to add labels of “generated images” and “error maps” to the upper and lower parts of Figure 4 for clearer presentation.
2. Is it possible to perform zero-shot generation on heterogeneous sensors (e.g., DIGIT)?

---

> ### Author Response · Authors · 2025-11-21
> **Response to Reviewer 3JNq (Part 1/2)**
>
> We sincerely appreciate your careful and thorough review of our manuscript. Below we provide point-by-point responses to your comments.
>
>  > Q1. In the downstream tasks, is ControlTac further fine-tuned, ...  I believe this is crucial to substantiating the paper’s claimed “cross-object generalization capability.”
>
> Thank the reviewer for the insightful comments. We would like to clarify the following regarding the generalization ability of ControlTac.
>
> ControlTac is not fine-tuned but directly utilizes the model trained on the FeelAnyForce Dataset (20,000 frames) in a zero-shot manner. To evaluate its generalizability, we have conducted additional experiments on two unseen objects, banana and nectarine, as presented in the manuscript. The results are included in ***Appendix D.2***, with example visualizations shown in ***Figures 4 and 17***.
>
> Table 1: Model performance on generating tactile images of unseen objects.
>
> | Object    | Model      | MSE  | SSIM |
> |-----------|------------|------|------|
> | Nectarine | ControlTac | 25   | 0.80 |
> | Nectarine | Hybrid     | 35   | 0.75 |
> | Nectarine | Separate   | 190  | 0.72 |
> | Nectarine | Taxim      | 978  | 0.69 |
> | Banana    | ControlTac | 27   | 0.78 |
> | Banana    | Hybrid     | 39   | 0.74 |
> | Banana    | Separate   | 207  | 0.72 |
> | Banana    | Taxim      | 1152 | 0.68 |
>
> Furthermore, we have validated its generalization ability to more complex objects and open environments through the pose estimation and object classification tasks, as discussed in ***Sections 4.3 and 4.4***. As shown in Table 2, for unseen objects such as the T-shape and Type-C connector, data generated by ControlTac can effectively train high-accuracy pose estimators. Additionally, as reported in Section 4.5, applying these estimators in real-world insertion tasks achieved success rates of 85% and 75%, further demonstrating ControlTac’s generalization capability.
>
> We acknowledge that performance is relatively lower for certain objects, such as bananas and rings; however, this issue can be mitigated by increasing the diversity of the training dataset. For example, after adding flat-surfaced cubes to the training set, the MSE for rings was reduced from 35 to 27, and the SSIM increased from 0.80 to 0.83, as discussed in Appendix E. This limitation is primarily due to the size and diversity of the current training data and the conditions used. Expanding the dataset and incorporating hardness and texture as additional conditions represent promising directions for future work.
>
>  > Q2. In the data generation process for downstream tasks, ... Could this potentially lead to contact mask–force pairs that are physically implausible?
>
> For the downstream tasks, the force range is set from 1 N to 10 N, and for each contact position, we generate 5 force values with a precision of 0.1 N, sampled randomly. This information has been added to ***Section 4.2***.
>
> We would like to emphasize that, as discussed in ***Section 3.1*** (Position-Control Mask) and the last paragraph of Section 4.1, our contact mask is used solely to determine the contact position and is independent of the contact force (as verified through detailed experiments in ***Appendix C***). Therefore, there is no risk of generating physically implausible contact mask–force pairs.
>
>  > Q3. The description of the object pushing ... Is the applied force static or dynamic during the pushing process?
>
> The objective of the object-pushing downstream task is to evaluate whether a force estimator trained on ControlTac-generated data can achieve performance comparable to one trained on real-world data, which further highlights the high quality of the data generated by ControlTac.
>
> In Table 7, two sets of numbers are provided for each object. For instance, in the entry "Bottle (0.63)", 0.63 refers to the mass of the glass bottle (0.63 kg). The numbers 1.08, 1.14, and 1.16 represent the measured pushing force: 1.08 N is the force measured by the force sensor, 1.14 N is the force predicted by the estimator trained on real-world data, and 1.16 N is the force predicted by the estimator trained on ControlTac-generated tactile data.
>
> During the pushing process, the applied force is dynamic. As the robotic arm pushes the object at a constant speed, we record the force sensor readings along with the corresponding tactile data, which is then used by the force estimator to predict the pushing force.
>
> We have added more details about this task in the ***Appendix D.6***.

---

> > ### Author Response · Authors · 2025-11-21
> > **Response to Reviewer 3JNq (Part 2/2)**
> >
> > > Q4. The real-world experiments have certain limitations, especially since the Insertion Task does not demonstrate clear real-time interaction characteristics. Can this work improve performance in more complex dexterous manipulation tasks?
> >
> > Currently, our pipeline is a purely tactile-controllable generation framework, which limits its direct use with robot-tactile paired data. As a result, it can only support estimation-based open-loop control for the insertion task.
> >
> > To improve inference efficiency, we adopt the diffusion transformer architecture from SANA, performing inference on a single NVIDIA RTX A6000 GPU with a batch size of 128. Under these settings, ControlTac achieves a throughput of 6.5 tactile images per second, demonstrating promising inference performance.
> >
> > As noted in the conclusion, a valuable future direction is to integrate our approach with robot data for training robot policies or dynamic models. In this context, our demonstrated capability for physics-conditioned realistic generation is particularly promising for augmenting robot datasets.
> >
> >  > Q5. It would be better to add labels of “generated images” and “error maps” to the upper and lower parts of Figure 4 for clearer presentation.
> >
> > Thanks for your suggestions. We have changed this figure and marked with blue text in the manuscript
> >
> >  > Q6.  Is it possible to perform zero-shot generation on heterogeneous sensors (e.g., DIGIT)?
> >
> > The current version can generalize to different samples of Gelsight Mini, as shown in the Sec. 4.3. However, it cannot do zero-shot generation on heterogeneous sensors because of the large difference of gels’ physical properties and color distributions. To this end, our framework has the potential to do this by scaling up the dataset with more sensors and adding more conditions about the gels’ properties via our ControlNet architecture. It can be a good extension.

---

> > > ### Comment · Reviewer_3JNq · 2025-11-27
> > >
> > > Thank you for your efforts in improving the paper and for your detailed responses. My concerns have been adequately addressed or discussed. Although the work still has some limitations, I believe its benefits to the tactile community outweigh them. Therefore, I will keep my original score and lean toward accepting the paper.

---

> > > > ### Author Response · Authors · 2025-11-27
> > > > **Response to Reviewer 3JNq**
> > > >
> > > > We sincerely thank the reviewer for the positive evaluation and recognition of our work. We appreciate the constructive comments and are glad that our revisions have addressed the concerns.

---

### Official Review · Reviewer_hxsZ · 2025-10-30

**Soundness:** 3
**Presentation:** 3
**Contribution:** 1
**Rating:** 2
**Confidence:** 2

**Summary:**

This work proposes to train a DiT+ControlNet to generate images corresponding to what a vision-based tactile sensor will see. The generation will be conditioned by a reference image from the real tactile sensor of the object of interest, the value of the force with which the sensor is touching the object and a contact map (a binary image) outlying the position of the object. By varying force and contact map the model can generate many visual tactile images from a single reference one better than what a simulator would do. The model can be used as data augmentation for real datasets and improve performances on 3 downstream tasks: force estimation, object pose estimation and object classification (among 6 classes). The specific models for each task are tested on offline datasets, but also in real condition with the use of a robotic harm. In general the data augmentation strategy based on the trained model is somewhat useful when the amount of available real data is small.

**Strengths:**

+ A quite interesting application of DiT+ControlNet to some under-explored touch. The way the authors designed the conditioning signals for their model makes a lot of sense in the context of the work and what they are trying to achieve.

+ According to Tab. 1 the method can recreate significantly more faithfully images from a tactile sensor for seen objects wrt to a separate simulator.

+ I appreciated that the authors took their proposal for a real world test including experiments with a real robot and various objects

**Weaknesses:**

## Major

A. **Experiments in Sec. 4.1 are all in-domain:** Per my understanding the results in Sec. 4.1. Cover only “in domain” experiments, meaning generations of images for known objects belonging to the same dataset used to train ControlTac. This puts the method and the ablations at an unfair advantage against Taxim that has not been fine tuned for that specific category of objects. The gap is big enough that the proposed method might still be better, but I would have expected a generalization experiment by removing from the training data a certain type of object and testing generation quality against it.

B. **Technical novelty:** While from an application perspective I feel like this paper checks all boxes, from a technical novelty perspective there is fundamentally limited innovation. The contribution is an application of DiT and ControlNet to this very specific robotic domain and showing that it can achieve promising results. There is not a strong technical novelty per se. I think it might be also a matter of the type of avenue where the work is submitted, I would find this work way more fitting for a robotic conference. I’m not sure what the broader ML community would find interesting from this work.


## Minor


C. **Minimal contributions to performance when enough real data are available:** Sec. 4.2 - Fig. 5 kinda shows that the learned model does not really add anything to a real dataset with enough samples. Note that the same real dataset is the one used to train ControlTac, so it is not very relevant to check performance when only a subset of them are available. Results seem more promising in Sec. 4.3 and Sec 4.4, but ties to Weakness A on the fact that without having a good measure of the generalization performance of the method it’s hard to say whether images generated with Taxim would have equally (or better) helped in these 2 experimental settings.

D. **Not very self-contained:** The paper presentation could be improved by providing a bit more background on tactile sensors and on what are the representations that are learned in the paper. The current draft assumes a significant amount of previous knowledge in the reader. I would suggest adding a section before 3.1 explaining what is a visual tactile sensor, what is the “background”, what is the “force vector” etc.

**Questions:**

1. What’s the impact of the choice of the reference image on the generation? If I pick 2 reference images from the same object and condition the generation with the same force and position control how much are the generated images changing?

2. Can you comment on weakness A?

---

> ### Author Response · Authors · 2025-11-21
> **Response to Reviewer hxsZ (Part 1/2)**
>
> We sincerely appreciate your careful and thorough review of our manuscript. Below we provide point-by-point responses to your comments.
>
> Response to the Major Concerns:
>
>  > A. Experiments in Sec 4.1 are all in-domain: ... but I would have expected a generalization experiment by removing from the training data a certain type of object and testing generation quality against it.
>
> We sincerely thank the reviewer for the insightful and constructive comments. We would like to provide some clarifications regarding our comparisons and experiments.
>
> First, Taxim is a physics-based simulator, and its sample-based physical parameter optimization is applied only for sensor calibration rather than for every object. Therefore, the comparison between our method and Taxim is fair.
>
> Second, although the objects themselves are the same, the contact positions used for generation are unseen in the training dataset. Even under these conditions, our method generalizes well to unseen objects such as T-shape, banana, and ring in downstream tasks like pose estimation and object classification.
>
> Third, we newly conduct an additional experiment in which the thin cylinder is removed, and the performance remains largely unchanged. The results are presented below and are also included in ***Appendix D.2***.
>
> Table 1: Effect of excluding the thin cylinder on model performance.
> | Dataset                 | MSE | SSIM |
> |-------------------------|-----|------|
> | Complete Dataset        | 23  | 0.83 |
> | Excluding Thin Cylinder | 25  | 0.82 |
>
> We also introduce a new evaluation on the generation of unseen objects. Specifically, we assess the performance of generating tactile images for bananas and nectarines. The results indicate that ControlTac performs robustly even on unseen objects. Detailed results are presented in the main text and ***Appendix D.2***, with example visualizations provided in ***Figures 4 and 17***.
>
> Table 2: Model performance on generating tactile images for unseen objects.
> | Unseen Object | MSE | SSIM |
> |---------------|-----|------|
> | Nectarine     | 25  | 0.80 |
> | Banana        | 27  | 0.78 |
>
>
>  > B. Technical novelty: While from an application perspective ... I’m not sure what the broader ML community would find interesting from this work.
>
> We appreciate the acknowledgment that our paper addresses all key aspects from an application perspective. We would like to note that ICLR currently features numerous ***robotics-related papers [1, 2, 3]*** as well as several ***tactile-related works [3, 4, 5]***, and we have chosen the ***“Applications to Robotics”*** track for our submission. Given your recognition of our contributions to the tactile domain in robotics, we hope you find our work valuable for this track.
>
> Additionally, we would like to emphasize the novelty of our application and framework design:
>
> Our framework is the first generative-based approach for tactile data augmentation in challenging downstream tasks, distinguishing it from prior generative models and outperforming physics-based simulators.
>
> To achieve this, we propose a specific and novel two-stage physical-conditioned ControlNet framework:
>  - We use a reference tactile image to extract features of the gel’s physical properties, optical variations, and object geometries.
>  - We incorporate physics variations as control inputs, which are specific to tactile images and differ from traditional visually-guided conditions such as sketches.
>  - We employ a multi-stage ControlNet backbone, enabling the model to adapt to varying amounts of data for different conditions.
>
> [1]. Lin et al., Data scaling laws in imitation learning for robotic manipulation, ICLR 2025
>
> [2]. Zheng et al., TraceVLA: Visual Trace Prompting Enhances Spatial-Temporal Awareness for Generalist Robotic Policies, ICLR 2025
>
> [3]. Liu et al., VTDexManip: A Dataset and Benchmark for Visual-tactile Pretraining and Dexterous Manipulation with Reinforcement Learning, ICLR 2025
>
> [4]. Gupta et al., Sensor-Invariant Tactile Representation, ICLR 2025
>
> [5]. Feng et al., AnyTouch: Learning Unified Static-Dynamic Representation across Multiple Visuo-tactile Sensors, ICLR 2025

---

> > ### Author Response · Authors · 2025-11-21
> > **Response to Reviewer hxsZ (Part 2/2)**
> >
> > Response to Minor Concern:
> >
> >  > A. Minimal contributions to performance when enough real data are available: ... Taxim would have equally (or better) helped in these 2 experimental settings.
> >
> > First, as highlighted in ***Appendix G.1***, the force estimation task is particularly challenging: a 0.5 N difference in force results in only subtle variations in the tactile images. Models trained on 13k real images outperform those trained on 14k or 15k images, indicating that the dataset is already sufficiently large for the model to approach near-oracle performance.
> >
> > Second, this sensitivity makes the task an excellent benchmark for evaluating the realism of our generated images. Even minor changes in force produce extremely subtle color differences, making the learning process highly sensitive. The key questions we aim to answer are: 1) Are the generated images realistic enough for this task? 2) What is the upper bound of the realism of our generated images? As explained in Section 4.2, our results show that the generated images are sufficiently realistic to achieve near-oracle performance using only half of the real data. In contrast, prior approaches using simulated or generated data cannot achieve this level of realism.
> >
> > Third, similar to our previous augmentations, although the objects themselves are the same, the contact positions used for generation are unseen. For other unseen objects, this experiment is not feasible due to insufficient real data for hybrid training. To evaluate generalization, we newly conduct additional experiments on unseen objects such as banana and nectarine, with the results shown in ***Table 2***, assessing the quality of the generated images.  The results indicate that ControlTac performs robustly even on unseen objects.
> >
> > In ***Sections 4.3 and 4.4***, some objects are entirely unseen, including USB Type-C connector, T-shape, banana, and ring. The results demonstrate that our pipeline generalizes well to these new objects. Notably, Section 4.3 highlights an important finding: models trained solely on real data perform worse than those trained on generated data. This is because it is practically impossible to collect real data covering all contact positions and angles on the sensor, whereas generated data can comprehensively cover these variations, leading to superior performance. An extreme example of this case is provided in Appendix G.2.
> >
> > > B. Not very self-contained: ... what is the “background”, what is the “force vector” etc.
> >
> > Thank you for the suggestion. We have newly added a subsection ***Sec. 3.1*** to provide more background on vision-based tactile sensing and force estimation from tactile sensors. . All changes are marked in blue.
> >
> > Response to Questions:
> >
> >  > Q2. What’s the impact of the choice of the reference image on the generation? ... position control how much are the generated images changing?
> >
> > We appreciate your suggestion regarding the impact of the reference image on generation results. Per your suggestion, we newly conduct additional experiments using ten different reference images of the same object under identical force and position control. The results show minimal variation across reference images, with 95% confidence intervals of MSE (22.66–23.34) and SSIM (0.8249–0.8331), indicating that the choice of reference image has negligible impact on the generated outputs. We have added these new results to ***Section 4.1***.

---

> > > ### Comment · Reviewer_hxsZ · 2025-11-26
> > > **Follow up on rebuttal**
> > >
> > > Thanks for addressing my criticisms and providing additional supporting experiments
> > >
> > > Re: Major Weakness A
> > >
> > > To complete the picture could you provide results for the competitors mentioned in Tab. 1 also for the Nectarine and Banana objects? It would help to measure whether the same performance advantage over existing methods holds also for unseen objects.
> > >
> > > Re: Major weakness B
> > >
> > > Acknowledge and I would not use the topic as a motivation to suggest rejection. Still some of the other works try to tackle more general problems that exist also for touch sensors. Like AnyTouch and sensor invariant tactile representation are about creating unified representation for different training data distributions (i.e., sensors) within one model, while VTDexManip spent some time discussing the influence of different modalities for the specific tasks they are benchmarking. From that point of view I found them more palatable for a broader ML community compared to this work that is a carefully designed data augmentation pipeline for touch sensor imagery. I agree that this is somewhat subjective and flagging it as a major weakness was maybe excessive. I will not weigh it too much in my final assessment. The point of technical novelty stands.
> > >
> > > Re: Weakness C and D
> > >
> > > Thanks for the additional pointers, I think these are mostly covered.

---

> ### Author Response · Authors · 2025-11-26
> **Response to Reviewer hxsZ**
>
> > Major Weakness A
>
> We sincerely thank the reviewer for pointing out this issue. We agree with your suggestion and have added a comparison of ControlTac with the baseline and simulator models on the Nectarine and Banana objects. As shown in the table, ControlTac consistently outperforms the other methods across both objects in terms of MSE and SSIM.
> The results are summarized in the table below. This content has also been included in **Appendix D.2, Table 7**.
>
> Table 1: Model performance on generating tactile images for unseen objects.
> | Object    | Model      | MSE  | SSIM |
> |-----------|------------|------|------|
> | Nectarine | ControlTac | 25   | 0.80 |
> | Nectarine | Hybrid     | 35   | 0.75 |
> | Nectarine | Separate   | 190  | 0.72 |
> | Nectarine | Taxim      | 978  | 0.69 |
> | Banana    | ControlTac | 27   | 0.78 |
> | Banana    | Hybrid     | 39   | 0.74 |
> | Banana    | Separate   | 207  | 0.72 |
> | Banana    | Taxim      | 1152 | 0.68 |
>
> > Major Weakness B
>
> We sincerely thank the reviewer for the understanding and for revisiting the earlier concern. We truly appreciate the reviewer’s thoughtful clarification.
>
> Indeed, our main contribution lies in addressing the shortage of high-quality tactile data, which remains a critical challenge for the tactile research community. As Reviewer Pyaz noted, “lack of high-quality tactile images is a bottleneck to the community.” For this challenges, we want to highlight a prior work ***Difftactile***, which is also accepted by ICLR, also tried to resolve similar ***tactile data scarcity*** problem using ***training-free simulator***. To the best of our knowledge, we are the first to leverage a generative model to produce higher-quality tactile data that can be directly used for downstream robot manipulation tasks—overcoming the limitations of previous free-form tactile generation methods, which could only generate lower-quality data suitable for perception tasks (e.g., object classification), as well as the significant sim-to-real gap in simulation-based approaches.
>
> We sincerely thank the reviewers for recognizing the contributions of our work.
>
> [1]. Si et al., DiffTactile: A Physics-based Differentiable Tactile Simulator for Contact-rich Robotic Manipulation, ICLR 2025

---

### Official Review · Reviewer_Pyaz · 2025-11-01

**Soundness:** 2
**Presentation:** 3
**Contribution:** 2
**Rating:** 4
**Confidence:** 4

**Summary:**

This paper proposed a tactile data augmentation method based on a single reference image called ControlTac.
In the first stage, authors applied a DiT that takes tactile image representation from a pre-trained SANA encoder and a 3D force vector as condition, to generate a corresponding target tactile image that reflects the desired force.
In the second stage, they reuse the DiT to allow the decoder to generate the same object under a 2D affine transformation.
With this pipeline, the authors could perform effective data augmentation.
Results have shown that this augmentation helps downstream offline perception and real-world manipulation experiments.

**Strengths:**

1. It is true that the lack of high-quality tactile images is a bottleneck to the community. A reliable augmentation method will benefit the tactile research community.
2. The authors designed and conducted extensive downstream experiments, providing qualitative and quantitative results to show the effectiveness of their method.

**Weaknesses:**

1. Fig 1 is not clear. From the context, I infer the "Data Augmentation" block is saying "generate tactile images for the same object with different 3D force and contact pose", but this invariance is not mentioned here or in the related context.
2. Regarding Fig 2 and related discussion: Text2Tac and Vis2Tac are obviously NOT tactile data augmentation methods. These two are not giving accurate tactile signals, but they try to align some tactile properties with other modalities in a generative model. It's inappropriate to put them in the context of this manuscript.
3. The definition of "tactile latent representation" is misleading for different settings. It's unclear when you mention the one from SANA, or after force injection.
4. It is still unclear why you use a contact mask to augment, given that we literally have the same object augmented with a 2D rotation. Shouldn't a 2D rotation matrix (3DoF) do the same job compared to a mask? When the object is larger than the sensor's size, how can masks help then? Also, how is edge detection involved in this stage?
5. On the motivation level, for a certain GelSight sensor, its deformation-to-force projection is determined by gel properties. This makes the cross-sensor claim questionable -- replacing background will only help when transferring across physically similar data with a bit of optical variance, limiting the actual contribution of this work.
6. Compared to simulation, I think one big advantage of your augmentation method is that you should be able to perform augmentation on objects with rich texture. However, only objects with simple geometry are shown in the figures. Is that due to the limitations of your encoder/decoder?

**Questions:**

See Weaknesses.

---

> ### Author Response · Authors · 2025-11-21
> **Response to Reviewer Pyaz (Part 1/2)**
>
> We sincerely appreciate your careful and thorough review of our manuscript. Below we provide point-by-point responses to your comments.
>
>  > Q1. Fig 1 is not clear. From the context, I infer the "Data Augmentation" block is saying "generate tactile images for the same object with different 3D force and contact pose", but this invariance is not mentioned here or in the related context.
>
> We appreciate the reviewer’s helpful feedback. We have clarified this point and updated ***Fig. 1*** accordingly. All changes are marked in blue.
>
>  > Q2. Regarding Fig 2 and related discussion: ...  It's inappropriate to put them in the context of this manuscript.
>
> We sincerely thank the reviewer for the insightful comments and would like to offer the following clarifications.
>
> First, we would like to respectfully provide a different perspective. Text2Tac and Vis2Tac are also tactile image generation pipelines that use diffusion-based methods, taking text or images as input. Because these methods are similar to ours, we believe they need to be discussed in the context of this manuscript. The key difference is that our method conditions on a reference tactile image and physical properties, and employs a two-stage ControlNet structure, which enables our framework to be the first generative approach designed specifically for data augmentation in dynamic downstream tasks.
>
> Second, we would like to clarify that prior works have already been used for tactile data augmentation in relatively simple tasks. For example, in Touching a NeRF and TaRF, the authors use generated tactile images for tactile localization and object classification, which serve as data augmentation applications in computer vision settings. In TaRF, the authors also describe their method as a tool to provide missing tactile inputs in the Tactile Augmented Radiance Field.
>
> Finally, we would like to highlight that we discuss these approaches only in the introductory sections, but do not include them in our experimental comparisons, as shown in ***Figure 7*** of the Appendix. Although these methods are also generative models, they are not sufficiently realistic for more challenging downstream tasks and do not incorporate conditioning on forces or positions, which is crucial for our problem setting.
>
>  > Q3. The definition of "tactile latent representation" is misleading for different settings. It's unclear when you mention the one from SANA, or after force injection.
>
> Thank you for the comment. We have clarified this point and updated ***Section 3.2*** accordingly. All changes are marked in blue.
>
>  > Q4. It is still unclear why you use a contact mask to augment, ... how can masks help then? Also, how is edge detection involved in this stage?
>
> Firstly, for labeling the position of the object in the training set or the reference image during inference, it is much easier to localize and label the contact mask of the object than to localize the pose or depth map of the object’s contact surface. One demo is shown in the supplementary videos.
>
> Secondly, as mentioned in ***Section 3.3***, other options such as pose or depth map are not accurate or consistent when the forces change.
>
> Thirdly, the contact mask can roughly demonstrate the contact area of the generated image, which cannot be obtained from a simple contact pose.
>
> Finally, during inference, we only need to label the initial contact mask, and the other inputs for the generated images are then transformed via a 2D rotation matrix as the reviewer mentioned.
>
>  > Q5. On the motivation level, ... limiting the actual contribution of this work.
>
> We would like to thank the reviewer for the insightful comments.
>
> For generalization to new samples, our method uses background-free tactile images as reference inputs and employs contact masks as conditions, allowing it to effectively capture implicit features of physical properties, color/optical distributions, and object geometries. In the pose estimation task, using a new GelSight Mini with a different gel material, our method continues to perform robustly and even surpasses models trained solely on real data.
>
> To further achieve more explicit and broadly applicable generalization across different sensor types, exploring improved ways of extracting gel physical properties will be an important extension of this project. With the ControlNet framework, our method has the potential to incorporate additional physical properties as conditions, enabling better generalization to a wider range of sensor modalities, which will be a key focus of our future work.

---

> ### Author Response · Authors · 2025-11-21
> **Response to Reviewer Pyaz (Part 2/2)**
>
> > Q6. Compared to simulation, ... Is that due to the limitations of your encoder/decoder?
>
> We sincerely appreciate the reviewer’s insightful comments.
>
> First of all, in Sections 4.3 and 4.4, we would like to highlight that our method can generalize to objects with different shapes and textures in both pose estimation and classification tasks, such as T-shape, banana, and rings. Visualizations of these objects are provided in ***Fig. 20***.
>
> Secondly, using a larger dataset and incorporating additional conditions can further improve generalization to more challenging objects. As noted in the failure cases, performance on bananas and rings is slightly lower than expected; however, this issue can be mitigated by increasing the diversity of the training dataset. For example, after adding flat-surfaced cubes to the training set, the MSE for rings was reduced from 35 to 27, and the SSIM increased from 0.80 to 0.83, as discussed in ***Appendix E***. Expanding the dataset and including additional conditions, such as hardness and texture, represents a promising direction for future work. Our framework, which integrates ControlNet, is well-suited for incorporating new conditions into the current model to further enhance generalization.
>
> Per request, we also include new evaluations on the generation quality of unseen objects. Specifically, we evaluate the performance of generating tactile images for banana and nectarine, which show comparable MSE and SSIM values to seen objects. More details have been added to the paper, and the results are presented below. These evaluations are also included in ***Appendix D.2***, with example visualizations shown in ***Figures 4 and 17***.
>
> Table 1: Model performance on generating tactile images for unseen objects.
> | Object    | Model      | MSE  | SSIM |
> |-----------|------------|------|------|
> | Nectarine | ControlTac | 25   | 0.80 |
> | Nectarine | Hybrid     | 35   | 0.75 |
> | Nectarine | Separate   | 190  | 0.72 |
> | Nectarine | Taxim      | 978  | 0.69 |
> | Banana    | ControlTac | 27   | 0.78 |
> | Banana    | Hybrid     | 39   | 0.74 |
> | Banana    | Separate   | 207  | 0.72 |
> | Banana    | Taxim      | 1152 | 0.68 |

---

### Author Response · Authors · 2025-11-21
**Shared Response to All Reviewers**

Dear all the reviewers:

Thanks for all the comments and insightful suggestions from reviewers! We also sincerely appreciate the reviewer for acknowledging the novelty and the empirical evaluation of our work – "a reliable augmentation method will benefit the tactile research community", "the authors designed and conducted extensive downstream experiments", "A quite interesting application of DiT+ControlNet to some under-explored touch", "a real world test including experiments with a real robot and various objects", “ControlTac can generate thousands of realistic tactile images from just one reference image”, and “the idea of physically grounded, force- and position-conditioned tactile generation is new”

First, we note that one of the main reasons for reviewer ***hxsZ***'s initial rating was the concern that ''this paper checks all boxes from an application perspective, but has limited technical contribution, making it more suitable for a robotics conference.'' We would like to give some justification:

We appreciate the acknowledgment that our paper addresses all key aspects from an application perspective. We would like to note that ICLR currently features numerous ***robotics-related papers [1, 2, 3]*** as well as several ***tactile-related works [3, 4, 5]***, and we have chosen the ***“Applications to Robotics”*** track for our submission. Given your recognition of our contributions to the tactile domain in robotics, we hope you find our work valuable for this track.

Then, we conduct some more experiments and update a new manuscript, and all changes are updated in blue:

 - Reply to reviewer ***Pyaz***, ***hxsZ***, and ***YcuJ***, we added new generation quality evaluation to evaluate the generalizability of our model. We evaluated the generation quality to two unseen objects, which are banana and nectarine, to show the generated images are still pretty realistic.

 - Reply to reviewer ***hxsZ***, we added a new experiment to show that the generation quality will not be influenced by using different reference images.

 - Reply to reviewer ***YcuJ***, we added new visualization for showing the gels’ deformation is also accurate by using depth maps.

 - Reply to reviewer ***YcuJ***, we tested the inference time and see that our model can generate 6.5 images every second, which is much faster than collecting real data.

 - Reply to reviewer ***hxsZ***, we added a new section 3.1 for showing backgrounds about vision-based tactile sensing and its contact forces.

Again, we thank the reviewers for their constructive feedback. We believe that all individual comments have been addressed in the respective responses, but we are happy to address any further comments from reviewers.

[1]. Lin et al., Data scaling laws in imitation learning for robotic manipulation, ICLR 2025

[2]. Zheng et al., TraceVLA: Visual Trace Prompting Enhances Spatial-Temporal Awareness for Generalist Robotic Policies, ICLR 2025

[3]. Liu et al., VTDexManip: A Dataset and Benchmark for Visual-tactile Pretraining and Dexterous Manipulation with Reinforcement Learning, ICLR 2025

[4]. Gupta et al., Sensor-Invariant Tactile Representation, ICLR 2025

[5]. Feng et al., AnyTouch: Learning Unified Static-Dynamic Representation across Multiple Visuo-tactile Sensors, ICLR 2025

---

### Author Response · Authors · 2025-12-02
**Summary of Rebuttal**

Dear Area Chair and Reviewers,

We would like to express our sincere gratitude to the reviewers and the Area Chair for the time and effort in evaluating our work, as well as for the valuable feedback that helped improve the work. We also sincerely appreciate the reviewer for acknowledging the novelty and the empirical evaluation of our work – "***will benefit the tactile research community***" (Reviewer Pyaz), "***extensive downstream experiments***" (Reviewer Pyaz), "A quite interesting application of DiT+ControlNet" (Reviewer hxsZ), “ControlTac can ***generate thousands of realistic tactile images from just one reference image***” (Reviewer 3JNq), and “the idea of physically grounded, force- and position-conditioned tactile generation is new” (Reviewer YcuJ).

During the discussion period, we provided detailed responses with additional experiments covering all the concerns. All the updated experimental results and analysis have been ***highlighted in blue*** in the manuscript. For more details about the updates in manuscript, please refer to our ***Shared Response***. Also, we would like to summarize our discussions with each reviewer:

Discussed with Reviewer ***3JNq***, the reviewer mentioned that "***all concerns have been adequately addressed or discussed***". Then, the reviewer chose to keep the ***original positive score***, raised the ***confidence from 4 to 5***, and lean toward accepting the paper.

Discussed with Reviewer ***hxsZ***, who initially gave the lowest rating. The reviewer has confirmed that most of the original concerns have been addressed, and we have resolved all remaining questions raised in the ongoing discussion:
 - We ***addressed the major concern*** about "Technical novelty". The reviewer ***acknowledged and would not use the topic as a motivation to suggest rejection***, acknowledged ***the subjective nature of this comment with respect to the venue of publication for this work***, and highlighted ***"will not weigh it too much in my final assessment"***.  According to the the initial comment, the review mentioned "from an application perspective I feel like this paper checks all boxes". For this point, we would like to highlight a prior work ***Difftactile [1]***, which ***accepted by ICLR 2025***, also tried to resolve similar ***tactile data scarcity*** problem using ***training-free simulator***.
 - We then conducted new comparisons with ***unseen objects*** "banana and nectarine", which addressed the concerns about the generalizabilities of our framework. ***(Sec. 4.1 and Sec. D.2)***
 - We also added a new section to show the background of our problem statement. ***(Sec. 3.1)***

Replied to Reviewer ***Pyaz***, we have addressed the following concerns:
 - We provided detailed analysis to highlight the difference between our method and previous generative-based methods.
 - We added more details about the choices about the contact mask.
 - We highlighted that our methods already have some ***cross-sample generation capabilities*** evaluated in our experiments ***(Sec. 4.2)***
 - We conducted new experiments with ***unseen objects*** "banana and nectarine" to highlight its generalizability across objects ***(Sec. 4.1)***.

Replied to Reviewer ***YcuJ***, we appreciate the ***original positive score*** and have addressed all following concerns:
 - We conducted new experiments with ***unseen objects*** "banana and nectarine" to highlight its generalizability across objects. ***(Sec. 4.1)***
 - We visualized the depth map to highlight that the generated images is ***physically grounded*** but not merely scale overall brightness. ***(Sec. G. 5)***
 - We conducted detailed analysis about the choices of baselines.
 - We reported the inference time and see that our model can generate 6.5 images every second, which is much faster than collecting real data and is valuable to accelerate the data scaling process. ***(Sec. A.2)***

Finally, we believe this work offers distinct contributions to the Tactile Sensing and Robot Learning Community:
 - ControlTac is the first framework that can generate ***physical-conditioned realistic tactile images***, which has potential to resolve the ***data scarcity*** in tactile sensing and benefit the community.
 - ControlTac is the first framework that can do data augmentation for ***dynamic manipulation-level tactile images***, which can benefit the robot learning community,

The additional experiments and analyses above further strengthen the paper. We sincerely thank the Reviewers and Area Chair for their time, effort, and constructive feedback.

Best regards,

The Authors of Submission 13062

[1]. Si et al., DiffTactile: A Physics-based Differentiable Tactile Simulator for Contact-rich Robotic Manipulation, ICLR 2025

---

### Meta-Review · Area_Chair_avcp · 2026-01-06

**Summary:**

The main concerns are:

Clarity of figures, method, motivation, and experiment.

Wrong statements like Text2Tac and Vis2Tac are obviously not tactile data augmentation methods.

Misleading definition: tactile latent representation.

Experiments are all in-domain.

The method is fundamentally limited innovation.

Physical faithfulness of force conditioning.

**Reviewer Concerns:**

Some of the concerns were addressed, e.g., missing results and discussion, and whether the baselines are proper or not. However,  the concerns about the novelty, clarity problem, and method design remained.

**Reviewer Scores:**

I believe the two negative reviewers would not change their minds to increase their scores.

---

### Decision · Program_Chairs · 2026-01-26

Reject